# Adaptive Gray: Reducing Color Dependency to Improve Generalization in Deepfake Detection

## Abstract

Deepfake technology, powered by advanced generative models such as GANs and diffusion models, has raised serious ethical and security concerns due to its potential for creating realistic yet deceptive content. As these generative models become increasingly sophisticated, humans struggle to distinguish real images from synthetic ones, highlighting the need for reliable machine-based detection. However, current detection methods often generalize poorly, especially across different generative models (cross-generator) and diverse image domains (cross-dataset), which limits their reliability in real-world deployment.

To address this issue, we observe that strong *color dependency* can be unnecessary and may even impede deepfake detection. Building on this insight, we propose Adaptive Gray (AG), a lightweight, learnable RGB-to-grayscale module that compresses color channels before classification to improve generalization. We validate AG on DiffusionForensics (in-distribution and cross-generator) and GenImage (cross-dataset) benchmarks. On GenImage, AG improves mean ACC / AP / TPR@5%FPR by over 15, 30, and 30 percentage points, respectively, compared to an RGB ResNet-50 baseline, while remaining competitive with or superior to strong state-of-the-art detectors. At the same time, AG introduces only three additional parameters and can reduce inference cost by up to $10^4\times$ relative to reconstruction-based pipelines, making it both effective and highly efficient for practical deepfake detection.

## 1 Introduction

The rapid proliferation of hyper-realistic images generated by advanced models like diffusion models Dhariwal & Nichol (2021); Ho et al. (2020); Nichol & Dhariwal (2021); Rombach et al. (2021); Mid (2022) poses growing societal threats, as humans increasingly struggle to distinguish synthetic content from genuine material Frank et al. (2024). Regulatory responses are emerging accordingly; for example, in September 2025 New South Wales (Australia) amended the *Crimes Act 1900* to criminalise the creation and sharing of sexually explicit deepfakes depicting real persons, with penalties up to three years' imprisonment (NSW, 2025). This highlights an urgent need for robust automated deepfake detection.

Despite significant research, current State-of-the-Art (SOTA) deepfake detection methods often fail to generalize effectively across unseen generative models (cross-generator) or diverse image domains (cross-dataset, e.g., faces vs. landscapes) Sha et al. (2023); Tan et al. (2024b); Wang et al. (2023). Our in-depth analysis reveals that while some SOTA approaches show promise in constrained settings, their performance in real-world, dynamic scenarios remains unsatisfactory. This critical limitation stems from their failure to capture intrinsic, robust distinctions between real and synthetic images.

**Our Work.** We identify a crucial and often overlooked bottleneck hindering deepfake detection generalization: the detrimental impact of an over-reliance on color dependency. Our systematic empirical analysis reveals that conventional RGB representations, despite their information richness, introduce excessive color-dependent redundancy and noise. This noise can mislead classifiers, causing them to learn spurious correlations rather than robust, intrinsic generative artifacts. Con-

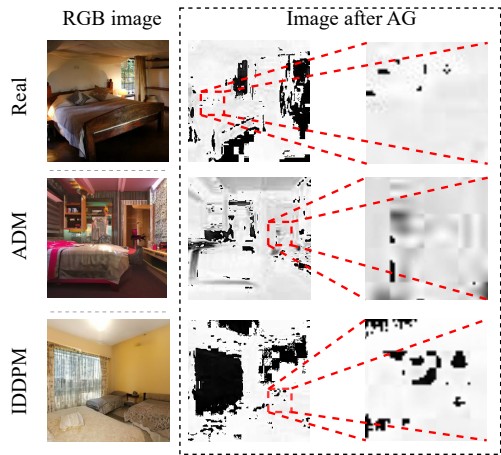

Figure 1: Qualitative comparison of artifacts in generated images before and after AG processing. The left column shows original RGB images, while the right column displays the same images after our proposed Adaptive Gray (AG) transformation. Notably, AG processing significantly enhances the visibility of texture-based artifacts inherent in generated images (e.g., ADM Dhariwal & Nichol (2021), IDDPM Nichol & Dhariwal (2021)) while suppressing color-dependent noise. This visual transformation makes distinguishing features more apparent and substantially improves classifier separation ability, particularly across diverse generative models.

sequently, models trained on full-color images struggle to adapt when color statistics shift across diverse datasets or novel generative models. Our key insight is that intelligently collapsing these color channels based on their intrinsic statistical differences can amplify hidden generative artifacts and reveal more general distinguishing features.

Inspired by this pivotal insight, we propose *Adaptive Gray* (AG), a novel, lightweight, and interpretable method. AG is designed to mitigate problematic color dependency and enhance generalization by intelligently transforming RGB images into a specialized grayscale representation. Unlike fixed grayscale conversions, AG learns optimal, data-driven coefficients to linearly combine RGB channels through an adaptive training process. This allows AG to effectively suppress irrelevant color details and accentuate subtle, texture-based artifacts unique to synthetic images, as visually demonstrated in Figure 1.

We conduct extensive experiments demonstrating AG's superior effectiveness and efficiency. Evaluated on challenging benchmark datasets like GenImage Zhu et al. (2024), AG consistently outperforms SOTA classifiers, achieving remarkable average improvements of 19.9% ACC, 22.0% AP, and 20.1% TPR@FPR=5%. Furthermore, AG significantly boosts inference efficiency by over $1 \times 10^4$ times. Our work not only highlights a fundamental challenge in deepfake detection but also offers a powerful, generalizable solution validated by substantial empirical gains.

**Contributions.** Our main contributions are summarized as follows:

- We uncover and systematically analyze a critical, overlooked factor affecting deepfake detection generalization: the detrimental impact of color dependency.

- Inspired by this insight, we propose *Adaptive Gray (AG)*, a novel, lightweight, and interpretable method designed to mitigate color dependency by learning optimal grayscale transformations.

- We conduct extensive experiments demonstrating AG's superior effectiveness, robustness, generalization, and remarkable inference efficiency across challenging datasets, reinforcing our initial hypothesis.

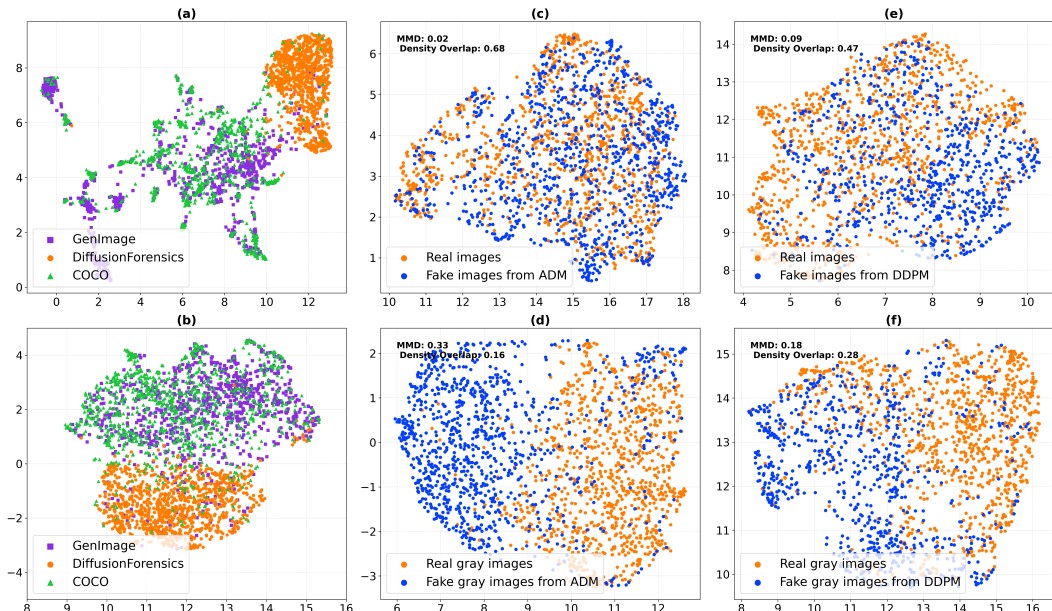

Figure 2: Distribution visualization before and after grayscale processing. Panels (a), (c) and (e) use RGB features, while (b), (d) and (f) show the same data after applying a fixed BT.601 grayscale projection (OG). (a)–(b) plot UMAP embeddings of three datasets (GenImage, DiffusionForensics, COCO), illustrating how grayscale processing changes the relative dataset distributions. (c)–(d) visualize real vs. ADM-generated images, and (e)–(f) visualize real vs. DDPM-generated images, again before and after grayscale. Each panel is annotated with MMD and Density Overlap scores to quantify the separation between real and fake samples.

## 2 RELATED WORKS

The detection of generated images has been widely explored in recent years McCloskey & Albright (2018; 2019); Guo et al. (2018). As GANs and diffusion-based models have advanced, distinguishing synthetic images from real ones has become increasingly difficult Karras et al. (2019); Brock (2018); Sauer et al. (2021).

Most existing methods focus on spatial artifacts introduced by generative models Yu et al. (2019); Marra et al. (2019); Ricker et al. (2024); Wang et al. (2020); Sarkar et al. (2024); Wang et al. (2023); Sha et al. (2023). For example, Wang et al. (2020) utilized deep neural networks to learn the distinguishing artifacts in generated images. However, their approach heavily relied on large amounts of GAN-generated images from diverse datasets. Sarkar et al. (2024) conducted a noteworthy study focusing on the coherence of lines and shadows in images, providing insights into how physical inconsistencies could reveal generated content. DIRE Wang et al. (2023) used reconstruction residuals for detection, but its reliance on expensive image reconstruction made it impractical for large-scale detection tasks.

For fake face detection, researchers focus on identifying some representative artifacts based on the distinct features of facial images Rossler et al. (2019); Haliassos et al. (2021); Wang & Deng (2021). For example, Haliassos et al. (2021) directed their work towards specific facial regions, such as the eyes and mouth, enhancing generated face detection by focusing on these distinctive areas. Wang & Deng (2021) improved fake face detection by employing an attention-based data augmentation method to guide the detector to explore representative facial regions.

Although these studies in deepfake detection have yielded promising results when training and testing on fake images from the same distribution, they still struggle to generalize detectors to fake images from different distributions, whether generated by different models or sourced from different datasets. For example, Jeong et al. (2022) strengthened model performance by generating unique fingerprints for classification. Chen et al. (2022) adopted adversarial training methods, enhanc-

ing model generalization and detection accuracy. Tan et al. (2024b) exploited upsampling artifacts through Neighboring Pixel Relationships (NPR), but their method struggled with datasets featuring diverse image distributions, as shown by our own experiments. While these approaches attempt to improve the generalization of deepfake detection for cross-model generalization, none of them focus on the challenge of cross-dataset generalization in deepfake detection.

Unlike previous methods, our method uniquely focuses on reducing color dependency in fake image detection, which allows us to avoid the need for computing new data representation features during inference, as required by Wang et al. (2023) and Tan et al. (2024b), thereby improving efficiency in the inference. Additionally, this streamlined approach significantly enhances the model's generalization capability (explained in Section 3.1).

## 3 METHODOLOGY

In this section, we introduce *Adaptive Gray (AG)*, a novel approach designed to enhance the generalization capabilities of deepfake detection models. Our methodology is fundamentally driven by a critical empirical observation: the often-detrimental impact of color dependency on model performance across diverse generative models and datasets. We begin by detailing the systematic analysis that led to this core insight, followed by the technical design of AG, which is engineered to effectively mitigate this issue through an adaptive, data-driven grayscale transformation process. Finally, we elaborate on the co-adaptive training strategy that enables AG to learn optimal image representations for robust deepfake detection. In practice, AG is implemented as a tiny learnable $1 \times 1$ projection over RGB channels, adding only three scalar parameters and thus incurring negligible overhead.

### 3.1 EMPIRICAL EVIDENCE FOR COLOR DEPENDENCY

Our work is premised on the hypothesis that an *unnecessary reliance on color information within deepfake detection models significantly hinders their generalization capability*. To rigorously validate this, we conducted a series of empirical analyses, focusing on how a simplified, grayscale representation of images affects key aspects of deepfake detection performance. Specifically, we aimed to address two core propositions:

- **Hypothesis 1 (H1):** Removing color dependency, through grayscale processing, will *reduce the intrinsic variability between deepfake datasets* (i.e., real and synthetic images from different sources) and *enhance the separability* between real and generated images within these datasets, thereby improving *cross-dataset generalization*.
- **Hypothesis 2 (H2):** Removing color dependency will *improve the separability of real and generated images* across various individual generative models, leading to better *cross-generator generalization*.

To conduct this foundational analysis, we employed a standard grayscale conversion based on the BT.601 luminance formula as a control mechanism. For an input RGB image $\mathbf{x} = [\mathbf{x}^{(R)}, \mathbf{x}^{(G)}, \mathbf{x}^{(B)}]$, where $\mathbf{x}^{(R)}$, $\mathbf{x}^{(G)}$, and $\mathbf{x}^{(B)}$ represent the red, green, and blue channels respectively, the grayscale image $\mathbf{x}_{\text{gray}}$ is computed as:

$$\mathbf{x}_{\text{gray}} = 0.299 \cdot \mathbf{x}^{(R)} + 0.587 \cdot \mathbf{x}^{(G)} + 0.114 \cdot \mathbf{x}^{(B)}. \tag{1}$$

This initial grayscale conversion serves to isolate the impact of color information, allowing us to observe if its removal fundamentally benefits deepfake distinction.

**Verifying Hypothesis 1 (H1).** To assess the impact on cross-dataset variability and separability, we analyzed the distributions of original RGB and grayscale images from three diverse datasets: DiffusionForensics Wang et al. (2023), GenImage Zhu et al. (2024), and COCO Lin et al. (2014). We utilized UMAP (Uniform Manifold Approximation and Projection) McInnes et al. (2018) to map high-dimensional image features into a two-dimensional space for visualization. As depicted in Figure 2 (a) and (b), our observations revealed that after grayscale processing, the feature distributions of these datasets became noticeably more aligned. This suggests that the elimination of color dependency reduces the inherent inter-dataset variability. More importantly, to quantify the separability between real and generated images, we employed two metrics: Density Overlap Li et al.

(2021) and Maximum Mean Discrepancy (MMD) Cheng & Xie (2021). Lower density overlap and higher MMD values indicate better separation. Our results (illustrated in Figure 2 (c) and (d), corresponding to real/generated image separation) unequivocally demonstrated that grayscale images significantly increased the separation between real and generated image distributions. This finding strongly supports H1, indicating that color often introduces distracting information that impedes effective cross-dataset generalization.

**Verifying Hypothesis 2 (H2).** To investigate the effect on cross-generator generalization, we conducted similar experiments by analyzing image distributions from various generative models. Focusing on the DDPM Ho et al. (2020) generator as a representative example, we compared the UMAP distributions of real and generated images before and after grayscale processing. As shown in Figure 2 (e) and (f), grayscale conversion led to a more pronounced separation between real and generated image distributions for this specific generator, consistently indicated by lower Density Overlap and higher MMD scores. This consistent improvement across different generators further substantiates H2, highlighting that generative models leave distinct, non-color-dependent artifacts that become more salient when color information is removed. Intuitively, real images exhibit strong cross-channel correlations due to the camera imaging pipeline (demosaicing, color correction, compression), while synthetic images often contain local, channel-specific inconsistencies introduced by upsampling and denoising steps. Grayscale projection suppresses redundant color variation and makes these texture inconsistencies easier to exploit for detection.

Our empirical analysis strongly suggests that *excessive reliance on color dependency within deep-fake detection models significantly compromises their generalization capability.* Grayscale processing, by mitigating this dependency, reveals more generalizable distinguishing features between real and generated images, laying the groundwork for improved detection across diverse scenarios.

## 3.2 ADAPTIVE GRAY (AG) FRAMEWORK

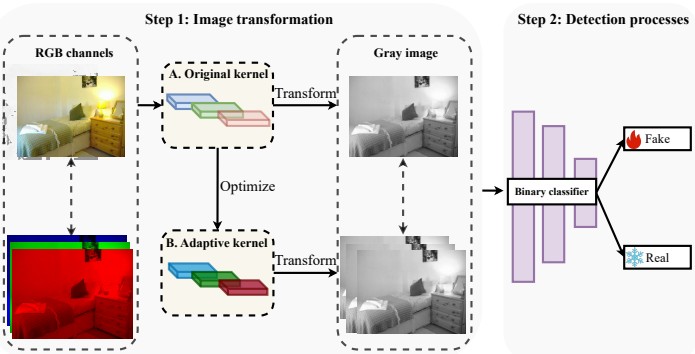

Figure 3: Pipeline of AG training. AG is initialized from the standard grayscale (OG) parameters and then refined together with the classifier. The frozen and fire icons indicate the alternating training scheme: in some epochs/cycles we freeze the AG parameters and update only the backbone, while in others we freeze the backbone and update only AG. In all stages we train on balanced batches of real and fake images.

Inspired by the empirical evidence that color dependency significantly impedes deepfake detection generalization, we propose *Adaptive Gray* (AG). Unlike conventional methods that rely on fixed color-to-grayscale conversion coefficients (as used in our empirical analysis), AG introduces a novel, data-driven approach to learn the optimal linear combination of RGB channels. This adaptive transformation is designed to effectively suppress misleading color information while accentuating the subtle, intrinsic textural artifacts that are highly indicative of synthetic origins.

The core idea of AG is to allow the deepfake classifier to actively participate in defining the most discriminative grayscale representation. As intuitively illustrated in Figure 1, AG processing amplifies these subtle texture-based artifacts that are typically obscured by color variations in generated images, thereby making the distinguishing features more salient for the classifier. The "Adaptive"

nature of our framework stems from an adaptive training process that jointly optimizes both the image transformation (grayscale conversion parameters) and the subsequent detection stages.

### 3.2.1 FORMALIZING THE ADAPTIVE GRAYSCALE TRANSFORMATION

We define the adaptive grayscale function, $\mathbf{G}(\mathbf{x}; \mathbf{w})$, which transforms an input RGB image $\mathbf{x} \in \mathbb{R}^{H \times W \times 3}$ (with height $H$, width $W$, and 3 color channels) into a single-channel grayscale image $\mathbf{x}' \in \mathbb{R}^{H \times W}$. This transformation is a linear combination of the input image's red, green, and blue channels:

$$\mathbf{x}' = \mathbf{G}(\mathbf{x}; \mathbf{w}) = w_R \cdot \mathbf{x}^{(R)} + w_G \cdot \mathbf{x}^{(G)} + w_B \cdot \mathbf{x}^{(B)}. \tag{2}$$

Here, $\mathbf{w} = [w_R, w_G, w_B]$ represents the learnable parameters (weights) for the grayscale transformation. Unlike the fixed coefficients in standard grayscale methods like BT.601 (Eq. 1), these parameters $w_R, w_G, w_B$ are not predetermined. Instead, they are dynamically optimized during the training process to best suit the deepfake detection task. The resulting image $\mathbf{x}'$ is what we refer to as the *Adaptive Gray* (AG) image. In our implementation, $w_R$, $w_G$ and $w_B$ are three scalar parameters shared across all spatial locations (equivalently, a $1 \times 1$ convolution over the RGB channels). This keeps AG extremely lightweight, adding only three degrees of freedom in front of an arbitrary backbone.

### 3.2.2 CO-ADAPTIVE TRAINING ALGORITHM

The effectiveness of AG lies in its co-adaptive training process, where the grayscale transformation parameters $\mathbf{w}$ and the deepfake binary classifier parameters $\theta$ are optimized in an alternating manner. This allows the grayscale conversion to adapt to the classifier's needs, and vice-versa, fostering a representation that maximizes separability between real and fake images. The training pipeline is illustrated in Figure 3, and the adaptive training process can be formally expressed as follows:

Let $L(\mathbf{y}, \mathbf{t})$ be the classification loss function (e.g., Binary Cross-Entropy), where $\mathbf{y}$ is the predicted probability from the classifier and $\mathbf{t}$ is the true binary label (real or fake). Let $\mathbf{f}(\cdot; \theta)$ denote the deepfake binary classifier with parameters $\theta$.

**Step 1: Optimize classifier parameters ($\theta$).** In this step, the AG parameters $\mathbf{w}$ are held fixed. The classifier $\mathbf{f}$ is then trained on the images already transformed by the current $\mathbf{w}$, with the objective of minimizing the classification loss:

$$\min_{\theta} L(\mathbf{f}(\mathbf{G}(\mathbf{x}; \mathbf{w}); \theta), \mathbf{t}). \tag{3}$$

This step enables the classifier to learn how to best classify images given the current adaptive grayscale representation, adapting its features to the information exposed by $\mathbf{G}(\mathbf{x}; \mathbf{w})$.

**Step 2: Optimize adaptive grayscale parameters ($\mathbf{w}$).** Subsequently, the classifier parameters $\theta$ are fixed. The AG parameters $\mathbf{w}$ are then optimized to further minimize the same classification loss, effectively learning how to transform the input images to make them most discriminative for the fixed classifier:

$$\min_{\mathbf{w}} L(\mathbf{f}(\mathbf{G}(\mathbf{x}; \mathbf{w}); \theta), \mathbf{t}). \tag{4}$$

This crucial step allows the AG weights to dynamically adjust their channel contributions, pushing the grayscale transformation towards a representation that highlights the most salient generative artifacts, thereby improving the overall model's generalization without requiring complex architectures or large additional parameters.

In practice, we alternate between these two steps over multiple training cycles. In each cycle, we first update the classifier for one epoch ($E_{\text{cls}} = 1$) with $\mathbf{w}$ frozen, and then update only $\mathbf{w}$ for one epoch ($E_{\text{ag}} = 1$) with $\theta$ frozen, **using the same optimizer and balanced batches of real and fake images in both stages**. We repeat this procedure for $K = 50$ cycles (100 epochs in total; see Sec. 4 for details). Empirically, this simple alternating scheme is more stable than naively optimizing $\mathbf{w}$ and $\theta$ jointly from scratch, and leads to better generalization; we provide an ablation in Sec. A.2 to support this design choice.

This alternating optimization process, typically performed over several training cycles, ensures that both the image representation and the classification model co-evolve, leading to a fine-tuned system that is highly effective and generalizable for deepfake detection.

## 4 EXPERIMENT SETUP

In this section, we introduce the experiment setup, including the datasets, models, baselines, metrics and implementation details.

**Train Datasets.** Following Wang et al. (2023), we use DiffusionForensics for training. Real images come from the *bedroom* category of LSUN (Yu et al., 2015). In total, the dataset contains 40,000 real images in the training set. Generated images are produced by ADM (Dhariwal & Nichol, 2021) in the same category (40k).

**Test Datasets.** Following previous studies Wang et al. (2023); Zhu et al. (2024); Tan et al. (2024b), we employed two types of test datasets in our study. (1) The first is the test dataset from *LSUN_bedroom* subset of the DiffusionForensics Wang et al. (2023) dataset Wang et al. (2023), which serves as an in-distribution dataset. This test set includes 1,000 real bedroom images, along with 1,000 generated images from each of the various generators trained on this dataset, including ADM Dhariwal & Nichol (2021), DDPM Ho et al. (2020), IDDPM Nichol & Dhariwal (2021), IF Saharia et al. (2022), Midjourney Mid (2022), PNDM Liu et al. (2022), ProGAN Karras et al. (2018), SD Rombach et al. (2021), and VQDM Gu et al. (2022). (2) The second is the GenImage dataset Zhu et al. (2024), used to evaluate cross-dataset generalization capability. GenImage Zhu et al. (2024) is specifically designed to assess the generalization performance of deepfake detection models and includes 6,000 real images and 6,000 high-quality generated images, covering 1,000 image categories for each of the generative models. In total, we consider eight generators from GenImage (ADM, BigGAN, Glide, Midjourney, SD-1.4, SD-1.5, VQDM and Wukong). The appendix provides the full eight-generator breakdown.

**Baselines.** We compare AG against several representative deepfake detectors: NPR Tan et al. (2024b), DIRE Wang et al. (2023), DE-FAKE Sha et al. (2023), UniversalFakeDetect (UniFD) Ojha et al. (2023), C2P-CLIP Tan et al. (2024a), D3 Yang et al. (2025) and FatFormer Liu et al. (2024). These methods span reconstruction-based detectors, frequency-aware CNNs, unified multi-generator models, discrepancy-based transformers, and CLIP-based architectures, and represent recent state-of-the-art baselines in synthetic image and deepfake detection. For fairness, all baselines in our experiments are trained and/or evaluated under a unified protocol unless otherwise specified: we use the same train/test splits, input resolution, and preprocessing as for our ResNet50 baseline, and report the same metrics (ACC, AP, and TPR@5%FPR). When official implementations or pretrained weights are available, we follow the authors' recommended hyperparameters as closely as possible; otherwise we re-train the models from scratch under this common setting. Further details are given in *Implementation details* and the Appendix.

**Metrics.** We selected evaluation metrics that align with previous research Wang et al. (2020); Tan et al. (2024b); Ricker et al. (2024); Qian et al. (2020); Sinitsa & Fried (2024) while considering real-world applicability. Common metrics such as *Average Precision (AP)* Wang et al. (2020); Tan et al. (2024b); Ricker et al. (2024) and *Average Accuracy (ACC)* Tan et al. (2024b); Qian et al. (2020); Sinitsa & Fried (2024) provide a comprehensive view of classifier performance. While AP evaluates performance across varying thresholds, real-world applications often lack prior knowledge of critical factors such as the true label of the image, the generative model used, and specific requirements (e.g., prioritizing either minimizing false positives or false negatives). As noted by Carlini Carlini et al. (2022) and other researchers Ho et al. (2017); Kantchelian et al. (2015); Kolter & Maloof (2006), evaluating models at lower False Positive Rates (FPR) provides a more realistic assessment. In line with Ricker Ricker et al. (2024), we included the *True Positive Rate (TPR)* at a fixed FPR, setting FPR to *5%* to balance sensitivity with the minimization of false positives, ensuring a practical threshold for generative image detection. *Threshold calibration and scoring.* We select a single global decision threshold on a held-out validation split from the **training domain** (LSUN_bedroom, ADM) to achieve **5% FPR**. This fixed threshold is then used to report TPR@5%FPR and ACC on all test sets (in-distribution, cross-generator, cross-dataset). AP is threshold-free.

**Implementation details.** We used ResNet50 He et al. (2016) as the backbone model for our classifier. We optimized the training process using the Adam optimizer with a learning rate of $10^{-4}$. The

batch size was set to 32, and training was conducted for 100 epochs, with the best model evaluated. For the AG variants, we adopt an alternating schedule with $E_{cls} = 1$, $E_{ag} = 1$ and $K = 50$ cycles (100 epochs in total), as described in Sec. 3.2.2. For image preprocessing, we resized images to $224 \times 224$ pixels *without additional data augmentation*. Unless otherwise noted, all methods use the same input preprocessing. Unlike prior work Wang et al. (2020), which shows that data augmentation can improve performance, we omitted it to directly assess the method's inherent generalization. All experiments were run on an Ubuntu server with an NVIDIA GeForce RTX 2080 Ti GPU.

## 5 EVALUATION

In this section, we present comprehensive experimental results that empirically validate our central hypothesis regarding the detrimental impact of color dependency on deepfake detection generalization, and demonstrate the superior efficacy of our Adaptive Gray (AG) method in addressing this challenge. We detail our experimental setup, followed by an in-depth analysis of AG's performance across various crucial aspects: in-distribution performance, cross-generator generalization, cross-dataset generalization, robustness to unseen perturbations, inference efficiency, an ablation study clarifying the role of adaptive learning, and a qualitative analysis.

### 5.1 IN-DISTRIBUTION PERFORMANCE

For fair evaluation, all models are trained and tested on in-distribution data from the same generative model (ADM Dhariwal & Nichol (2021)) and image category (LSUN_bedroom Yu et al. (2015)). As shown in Table **??**, recent baselines already achieve very strong in-distribution results: DIRE, UniFD, and D3 all exceed 95% ACC with AP/TPR very close to 100%, while transformer/CLIP-based C2P-CLIP and FatFormer further push ACC to about 99% with near-saturated AP. Our OG and AG variants match or slightly exceed these numbers, with OG achieving 99.9% ACC and essentially 100% AP/TPR, and AG reaching 99.9% ACC and 100.0% AP/TPR. This shows that collapsing RGB to a single (fixed or learned) gray channel does not sacrifice in-distribution accuracy, even in the presence of strong modern baselines, and establishes a solid foundation before we evaluate AG under the more challenging cross-generator (Sec. 5.2) and cross-dataset (Sec. 5.3) settings.

| Method | ACC | AP | TPR@5%FPR |
|---|---|---|---|
| ResNet | 90.1 | 93.3 | 96.7 |
| DIRE | 95.1 | 99.9 | 99.7 |
| NPR | 50.0 | 44.7 | 3.0 |
| DE-FAKE | 91.5 | 94.3 | 80.3 |
| UniFD | 95.4 | 99.9 | 99.3 |
| C2P-CLIP | 98.9 | 99.3 | 99.0 |
| FatFormer | 99.1 | 99.0 | 99.0 |
| D3 | 97.7 | 99.7 | 98.6 |
| OG | **99.9** | 99.9 | 99.9 |
| AG (Ours) | **99.9** | **100.0** | **100.0** |

Table 1: In-distribution performance on DiffusionForensics (ADM). **Bold** and underline indicate the best and the second-best performance, respectively.

### 5.2 CROSS-GENERATOR PERFORMANCE

We conduct cross-generator experiments to analyse generalization capabilities. Models are trained on ADM-generated and LSUN_bedroom real images, and then evaluated on the remaining generators in the DiffusionForensics dataset Wang et al. (2023) (fixed "bedroom" category). As summarized in Table 2, our OG variant, which applies a fixed grayscale transformation, already yields a substantial improvement over the RGB ResNet50 baseline, confirming that mitigating color dependency alone boosts transfer to novel generators. Building on this, AG further elevates performance and attains the best mean ACC, AP, and TPR across all compared methods on unseen generators (94.6% ACC, 99.5% AP, and 99.7% TPR), often reaching nearly perfect detection for individual generators. Compared to DIRE, AG matches its AP ($\approx$99.5%) while improving both ACC and TPR, and it also clearly surpasses more recent baselines such as UniFD, C2P-CLIP, FatFormer, D3, and ConvNeXt, which leverage large pre-trained models and transformer/CLIP-style architectures. In contrast, methods such as NPR and DE-FAKE show more pronounced fluctuations across generators. Taken together, these results support Hypothesis 2 (H2) from Sec. 3.1: explicitly reducing color dependency via (adaptive) grayscale projection is a key ingredient for robust cross-generator detection, and the learned AG kernel provides the most effective trade-off among all strong baselines.

| Method | ACC | AP | TPR@5%FPR |
|---|---|---|---|
| ResNet50 | 59.0 | 59.9 | 24.4 |
| ConvNeXt | 92.5 | 94.4 | 95.9 |
| ViT | 81.9 | 78.5 | 55.0 |
| DIRE | 94.1 | **99.5** | 98.0 |
| NPR | 51.3 | 48.9 | 4.3 |
| DE-FAKE | 78.0 | 76.5 | 43.4 |
| DIRE+AG | 92.8 | 97.5 | 90.5 |
| ConvNeXt+AG | 92.9 | 96.6 | 97.3 |
| ViT+AG | 84.9 | 85.1 | 61.1 |
| UniFD | 72.0 | 92.0 | 73.5 |
| C2P-CLIP | 80.0 | 89.0 | 72.9 |
| Fatformer | 83.1 | 93.7 | 83.4 |
| D3 | 94.4 | 97.0 | 92.9 |
| OG | 87.3 | 85.9 | 74.1 |
| AG (ours) | **94.6** | **99.5** | **99.7** |

Table 2: Cross-generator performance on the DiffusionForensics dataset. Values are averaged over these generators.

## 5.3 CROSS-DATASET PERFORMANCE

| Methods | ACC | AP | TPR@5%FPR |
|---|---|---|---|
| ResNet50 | 47.0 | 43.1 | 1.3 |
| ConvNeXt | 59.7 | 74.5 | 24.3 |
| ViT | 54.1 | 59.8 | 12.8 |
| DIRE | 54.5 | 75.0 | 22.7 |
| NPR | 50.0 | 47.1 | 5.5 |
| DE-FAKE | 59.4 | 65.4 | 20.4 |
| DIRE+AG | 58.7 | 67.2 | 0.0 |
| ConvNeXt+AG | 60.5 | 76.7 | 31.6 |
| ViT+AG | 56.8 | 60.4 | 13.7 |
| UniFD | 60.8 | 66.6 | 28.9 |
| C2P-CLIP | 60.4 | 68.6 | 31.0 |
| FatFormer | 50.1 | 50.1 | 1.6 |
| D3 | 62.2 | 70.6 | 29.3 |
| OG | 62.2 | 70.4 | 26.9 |
| AG (Ours) | **62.4** | **77.3** | **33.7** |

Table 3: Cross-dataset performance on GenImage. Numbers are averaged over generators.

Building on the cross-generator insights, we next evaluate cross-dataset generalization by training on DiffusionForensics (ADM, LSUN_bedroom) and testing on the GenImage benchmark Zhu et al. (2024). GenImage contains images from eight different generators (ADM, BigGAN, Glide, Midjourney, SD1.4, SD1.5, VQDM and Wukong) spanning 1,000 ImageNet-style classes. Table ?? reports mean performance across all eight generators, while the full results are given in Appendix A (Tables 8–10).

AG again achieves the best cross-dataset performance, with mean ACC of 62.4%, AP of 77.3% and TPR@5%FPR of 33.7%. Compared to DE-FAKE, AG improves mean ACC by +3.0% and TPR by +13.3%, and compared to DIRE it further increases AP by +2.3%, while OG already improves substantially over the vanilla ResNet50 backbone. Moreover, AG also outperforms more recent baselines such as UniFD, C2P-CLIP, D3 and ConvNeXt on mean ACC/AP/TPR across all eight generators. These trends are consistent across almost all generators in the appendix tables, confirming that mitigating color dependency via AG is beneficial under distribution shift.

In summary, AG consistently outperforms or remains highly competitive with other SOTA detection methods Wang et al. (2023); Tan et al. (2024b); Sha et al. (2023); Ojha et al. (2023); Tan et al. (2024a); Liu et al. (2024); Yang et al. (2025) across all testing scenarios—in-distribution, cross-generator, and especially the challenging cross-dataset conditions. These results not only demonstrate AG's superior practical utility but also provide strong empirical validation for our core hypothesis that mitigating color dependency is fundamental for achieving robust and generalizable deepfake detection.

## 5.4 ROBUSTNESS TO UNSEEN PERTURBATIONS

Robustness to common image perturbations is crucial for real-world deepfake detection, as images often undergo degradations. We evaluated AG against DIRE Wang et al. (2023), NPR Tan et al.

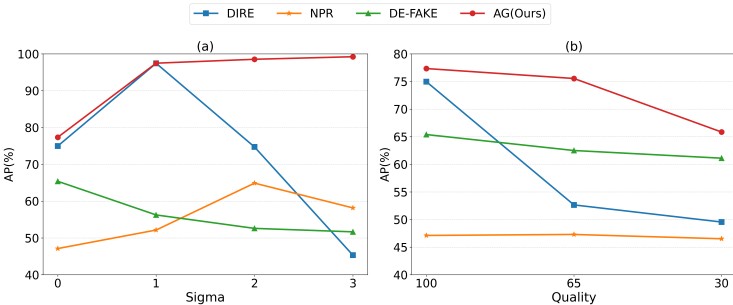

Figure 4: Robustness to unseen perturbations. Figure (a) illustrates performance against Gaussian blur, and Figure (b) shows performance against JPEG compression for AG, NPR, DE-FAKE, and DIRE. We evaluate the average AP of all methods on GenImage, using images from BigGAN, Midjourney, Stable Diffusion, and Wukong models.

(2024b), and DE-FAKE Sha et al. (2023) under Gaussian blur ($\sigma = 1, 2, 3$) and JPEG compression (quality 65, 30), following previous studies Wang et al. (2023).

As shown in Figure 4, AG consistently outperforms baselines across all blur and compression levels. For Gaussian blur, AG maintains high and stable Average Precision (AP). Under JPEG compression, AG exhibits significantly less performance degradation; specifically, while DIRE experiences a substantial AP drop at JPEG quality 30, AG remains remarkably resilient. This outcome reinforces our core hypothesis: AG's adaptive grayscale processing reduces reliance on superficial color information, making it inherently more resilient to common image degradations that often corrupt color channels. Its focus on robust, underlying texture-based artifacts is key to this enhanced robustness.

## 6 CONCLUSION

In this work, we tackled the challenge of enhancing binary deepfake classifiers' generalization by reducing color dependency in real vs. generated image detection. We proposed that color discrepancies can hinder detection accuracy, leading to the development of our grayscale processing framework, **Adaptive Gray (AG)**. Through adaptive training of both grayscale parameters and the classifier, AG demonstrated superior generalization across datasets and testing conditions, outperforming SOTA methods. Our findings suggest that focusing on texture over color can improve detection resilience, offering a promising direction for generalizable detection systems applications.

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

# A  APPENDIX

## A.1  EFFICIENCY EVALUATION

Table 4: Inference Time Comparison of Different Methods.

| Method | Average Inference Time per Image (s) |
|---|---|
| DIRE | 269.50 |
| DE-FAKE | 12.12 |
| NPR | $8.66 \times 10^{-3}$ |
| AG (Ours) | $\mathbf{6.12} \times 10^{-3}$ |

AG's efficiency stems from its simple, linear processing during inference. Once $\mathbf{w}$ is learned, image transformation involves only straightforward multiplication and addition, making AG comparable in cost to a standard ResNet50 classifier He et al. (2016). This contrasts sharply with DE-FAKE's multi-component pipeline (CLIP encoders, large classifier, optional BLIP for captioning) and DIRE's computationally intensive diffusion reconstruction, both of which are impractical for real-time or large-scale applications.

Combined with its strong generalization, AG's efficiency positions it as a practical solution for real-time deepfake detection and high-volume image analysis, alleviating common computational bottlenecks.

## A.2 Ablation Study: The Impact of Adaptive Grayscale Learning

Our methodology posits that both basic grayscale processing and adaptive learning of its coefficients enhance deepfake detection generalization. To dissect these contributions, we address two research questions (RQs) related to our empirical hypotheses (H1 and H2) from Section 3.1:

- **RQ1: Basic Grayscale Efficacy** Does fixed grayscale compression improve generalization by retaining relevant features?

- **RQ2: Value of Adaptive Learning:** Does optimizing grayscale parameters (AG) further enhance generalization beyond fixed grayscale, suggesting a more discriminative compression learned by machines?

To verify **RQ1**, we evaluated the Original Gray (OG) method, which uses standard BT.601 grayscale conversion (Eq. 1) without adaptive training. As shown in Table 2 and Table **??**, even fixed OG processing notably improves generalization. In cross-generator testing, OG increases ACC from 59.0% to 87.3% and AP from 59.9% to 85.9% over the baseline ResNet50 He et al. (2016). In cross-dataset testing on GenImage, OG still gains from 47.0% to 62.2% in ACC and from 43.1% to 70.4% in AP. These findings strongly support RQ1, confirming that simply mitigating color dependency through fixed grayscale conversion already enhances generalization by retaining critical texture-based features.

To verify **RQ2**, we compared OG and AG to see if optimizing grayscale parameters further enhances generalization. This addresses whether machine learning can discover a more optimal grayscale kernel. Results from Table 2 and Table **??** confirm that adaptively training AG parameters improves classifier generalization beyond fixed OG. In cross-generator testing, AG surpasses OG with ACC increasing from 87.3% to 94.6%, AP from 85.9% to 99.5%, and TPR@5%FPR from 74.1% to 99.7%. On GenImage, AG further improves AP (70.4%→77.3%) and TPR (26.9%→33.7%) while maintaining similar ACC (62.2%→62.4%). These results support RQ2, indicating that the co-adaptive training process enables AG to learn a more discriminative grayscale projection, especially for challenging cross-generator generalization.

## A.3 Exploratory Analysis: Integrating Adaptive Gray with DIRE and Modern Backbones

Beyond the ResNet50 backbone used in our main experiments, we also treat Adaptive Gray as a plug-and-play module and explore its effect when integrated into stronger architectures such as ConvNeXt and ViT. On DiffusionForensics cross-generator evaluation (Table 2), AG+ConvNeXt improves the mean ACC/AP/TPR from 92.5/94.4/95.9% (ConvNeXt) to 92.9/96.6/97.3%, while AG+ViT boosts ViT from 81.9/78.5/55.0% to 84.9/85.1/61.1%. On the cross-dataset GenImage benchmark (Tables 8–10), the same pattern holds: AG+ConvNeXt increases mean ACC/AP/TPR from 59.7/74.5/24.3% to 60.5/76.7/31.6%, and AG+ViT improves ViT from 54.1/59.8/12.8% to 56.8/60.4/13.7%. These consistent gains across CNN and transformer backbones support our claim that learning an adaptive grayscale kernel is complementary to modern discriminative architectures, rather than being specific to ResNet50.

However, the results in Table 2 and the GenImage tables (Tables 8–10) show that this combination does not improve generalization over AG alone. On DiffusionForensics, DIRE+AG lags behind AG in mean ACC, AP and TPR@5%FPR, and on GenImage its mean ACC/AP are lower while the TPR@5%FPR collapses to zero across generators.

We hypothesize that this is due to a form of over-compression. DIRE already operates on reconstruction residuals from a diffusion model, which remove much of the original image content (shape, texture, and color). Applying an additional grayscale compression step on top of these residuals likely discards too many informative cues, leading to unstable and degraded performance. Taken together with the positive AG+ConvNeXt and AG+ViT results above, this suggests that AG is most effective when applied directly to RGB inputs in front of standard discriminative backbones, whereas naively chaining AG with already heavily compressed residual signals (as in DIRE) can be detrimental.

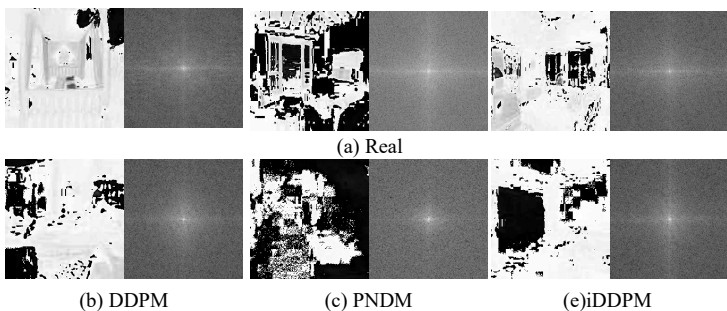

(a) Real

(b) DDPM  (c) PNDM  (e)iDDPM

Figure 5: Frequency-domain analysis using Fast Fourier Transform (FFT) on AG grayscale images, showcasing distinct characteristics between real and generated images. (a) Real images typically demonstrate a richer frequency spectrum with organized noise patterns aligned with natural object structures. In contrast, generated images from models such as DDPM and PNDM exhibit more irregular and artificial frequency distributions. This comparison highlights that FFT-based analysis on AG-transformed images effectively enhances the separability of real from generated content in deepfake detection, by making subtle generative artifacts more visually apparent in the frequency domain.

| Method | ADM | Dalle2 | DDPM | StyleGAN | IDDPM | IF | LDM | Midjourney | PNDM | ProGAN | SD2 | VQDM | Mean |
|---|---|---|---|---|---|---|---|---|---|---|---|---|---|
| ResNet50 | 90.0 | 66.7 | 56.6 | 50.0 | 50.0 | 50.0 | 50.0 | 94.9 | 50.0 | 50.0 | 50.0 | 50.0 | 59.0 |
| DIRE | 95.1 | 93.4 | 93.7 | 94.8 | 94.9 | 94.8 | 94.4 | 91.2 | 92.4 | 94.0 | 95.1 | 95.1 | 94.1 |
| NPR | 50.0 | 33.3 | 63.9 | 50.0 | 50.0 | 92.5 | 50.0 | 9.1 | 50.0 | 66.7 | 50.0 | 50.0 | 51.3 |
| DE-FAKE | 91.4 | 68.9 | 90.2 | 71.5 | 93.2 | 72.2 | 88.7 | 83.3 | 88.2 | 78.2 | 55.2 | 55.3 | 78.0 |
| UniFD | 95.4 | 89.3 | 73.7 | 79.4 | 73.4 | 50.0 | 50.8 | 53.0 | 86.2 | 81.8 | 53.4 | 77.8 | 72.0 |
| C2P-CLIP | 98.9 | 67.5 | 71.7 | 99.9 | 80.1 | 64.6 | 97.5 | 91.1 | 82.8 | 90.1 | 59.1 | 55.1 | 79.9 |
| FatFormer | 99.1 | 81.6 | 62.6 | 100.0 | 70.3 | 58.8 | 95.3 | 92.7 | 99.5 | 98.8 | 74.3 | 62.9 | 83.0 |
| D3 | 97.6 | 71.2 | 95.3 | 99.0 | 99.2 | 92.3 | 97.2 | 90.0 | 99.8 | 98.8 | 93.8 | 98.6 | 94.4 |
| DIRE+AG | 98.9 | 92.2 | 95.1 | 83.1 | 98.8 | 92.6 | 91.5 | 97.2 | 97.3 | 80.2 | 93.8 | 92.6 | 92.8 |
| ConvNeXt | 99.4 | 78.3 | 93.8 | 89.3 | 99.4 | 99.4 | 99.4 | 75.4 | 99.4 | 88.9 | 92.2 | 94.6 | 92.5 |
| AG+ConvNeXt | 99.2 | 83.1 | 97.8 | 84.0 | 98.2 | 97.2 | 99.2 | 90.7 | 95.1 | 82.8 | 92.5 | 95.4 | 92.9 |
| ViT | 93.1 | 62.8 | 92.4 | 81.9 | 94.0 | 72.9 | 87.6 | 82.4 | 91.4 | 84.4 | 64.9 | 75.0 | 81.9 |
| AG+ViT | 91.2 | 74.1 | 90.6 | 77.6 | 92.4 | 87.5 | 90.0 | 85.7 | 89.8 | 80.4 | 82.8 | 76.8 | 84.9 |
| OG | 100.0 | 73.7 | 99.3 | 93.1 | 99.4 | 66.6 | 98.5 | 90.1 | 99.0 | 85.7 | 57.2 | 85.0 | 87.3 |
| AG (ours) | 100.0 | 78.7 | 100.0 | 78.4 | 100.0 | 97.4 | 98.4 | 92.9 | 99.9 | 97.8 | 97.4 | 99.4 | 95.0 |

Table 5: Cross-generator accuracy (ACC, %) on DiffusionForensics. All models are trained on ADM (LSUN_bedroom) and evaluated on 12 generators.

## A.4 QUALITATIVE ANALYSIS OF AG

To further illuminate AG's underlying mechanisms and properties, we performed a qualitative frequency-domain analysis using Fast Fourier Transform (FFT) on AG-transformed images, following our quantitative results confirming AG's effectiveness through reduced color dependency. This analysis aimed to visually and analytically explain how AG enhances discriminative power.

As illustrated in Figure 5, AG processing reveals distinct low-level frequency characteristics. Real images, after AG transformation, exhibit a richer, more organized frequency spectrum with noise patterns aligning with natural structures. Conversely, generated images (e.g., ADM, DDPM) display irregular, chaotic, or unnatural frequency distributions. These irregularities are subtle generative artifacts that become significantly more pronounced and detectable in the grayscale frequency domain, often obscured by complex color patterns in RGB space. This strongly supports our hypothesis: AG effectively accentuates intrinsic, non-color-dependent artifacts for generalized deepfake detection.

## A.5 FULL RESULTS ON DIFFUSIONFORENSICS

## A.6 FULL RESULTS ON GENIMAGE

| Method | ADM | Dalle2 | DDPM | StyleGAN | IDDPM | IF | LDM | Midjourney | PNDM | ProGAN | SD2 | VQDM | Mean |
|---|---|---|---|---|---|---|---|---|---|---|---|---|---|
| ResNet50 | 93.3 | 58.0 | 48.8 | 54.9 | 51.1 | 38.7 | 67.9 | 98.0 | 42.6 | 54.9 | 69.3 | 41.6 | 59.9 |
| DIRE | 100.0 | 99.9 | 99.3 | 99.4 | 99.7 | 99.6 | 99.2 | 100.0 | 97.9 | 98.1 | 98.8 | 100.0 | 99.3 |
| NPR | 44.7 | 36.2 | 57.5 | 53.9 | 43.7 | 91.9 | 35.8 | 9.0 | 9.0 | 73.2 | 63.3 | 39.6 | 46.5 |
| DE-FAKE | 94.3 | 51.4 | 92.3 | 82.6 | 96.3 | 82.1 | 92.4 | 13.6 | 93.1 | 86.1 | 68.0 | 65.9 | 76.5 |
| UniFD | 99.9 | 99.5 | 97.6 | 98.0 | 96.7 | 59.8 | 86.0 | 72.6 | 99.1 | 98.8 | 84.8 | 99.0 | 91.0 |
| C2P-CLIP | 99.3 | 73.6 | 87.3 | 100.0 | 96.2 | 94.2 | 99.9 | 51.8 | 98.2 | 95.1 | 88.3 | 83.2 | 88.9 |
| FatFormer | 99.0 | 97.2 | 76.0 | 100.0 | 94.0 | 94.3 | 99.9 | 81.7 | 99.9 | 99.9 | 95.0 | 86.1 | 93.6 |
| D3 | 99.7 | 79.0 | 98.2 | 100.0 | 100.0 | 97.5 | 99.5 | 92.1 | 100.0 | 99.9 | 98.5 | 99.2 | 97.0 |
| DIRE+AG | 100.0 | 95.4 | 98.5 | 95.8 | 100.0 | 98.4 | 97.9 | 93.8 | 99.7 | 93.6 | 98.8 | 98.2 | 97.5 |
| ConvNeXt | 100.0 | 89.7 | 100.0 | 99.0 | 100.0 | 100.0 | 100.0 | 64.4 | 100.0 | 89.9 | 90.0 | 100.0 | 94.4 |
| AG+ConvNeXt | 100.0 | 86.0 | 100.0 | 99.6 | 100.0 | 99.5 | 100.0 | 86.6 | 99.9 | 99.3 | 88.6 | 99.7 | 96.6 |
| ViT | 87.9 | 32.3 | 96.4 | 91.0 | 98.4 | 84.1 | 94.6 | 6.0 | 95.1 | 92.6 | 77.9 | 86.0 | 78.5 |
| AG+ViT | 96.8 | 63.4 | 95.2 | 86.2 | 97.4 | 92.5 | 95.3 | 35.5 | 95.0 | 88.4 | 90.2 | 85.0 | 85.1 |
| OG | 100.0 | 64.9 | 100.0 | 98.8 | 100.0 | 88.2 | 99.9 | 6.1 | 100.0 | 98.3 | 77.0 | 97.9 | 85.9 |
| AG (ours) | 100.0 | 99.4 | 100.0 | 99.8 | 100.0 | 100.0 | 100.0 | 94.4 | 100.0 | 100.0 | 99.9 | 100.0 | 99.4 |

Table 6: Cross-generator average precision (AP, %) on DiffusionForensics.

| Method | ADM | Dalle2 | DDPM | StyleGAN | IDDPM | IF | LDM | Midjourney | PNDM | ProGAN | SD2 | VQDM | Mean |
|---|---|---|---|---|---|---|---|---|---|---|---|---|---|
| ResNet50 | 96.7 | 25.8 | 11.1 | 10.0 | 5.7 | 0.0 | 16.0 | 99.0 | 3.5 | 10.5 | 14.4 | 0.4 | 24.4 |
| DIRE | 99.7 | 99.9 | 98.6 | 99.3 | 99.2 | 99.5 | 96.0 | 98.6 | 89.7 | 95.9 | 99.9 | 99.9 | 98.0 |
| NPR | 3.0 | 5.0 | 2.0 | 7.3 | 1.9 | 5.2 | 1.9 | 5.2 | 5.2 | 11.0 | 1.0 | 1.0 | 4.1 |
| DE-FAKE | 80.3 | 6.8 | 77.7 | 29.2 | 100.0 | 28.5 | 64.5 | 2.0 | 73.1 | 41.7 | 7.7 | 9.0 | 43.4 |
| UniFD | 99.3 | 97.0 | 85.0 | 88.8 | 82.3 | 10.0 | 45.5 | 47.0 | 94.4 | 93.2 | 43.2 | 95.7 | 73.5 |
| C2P-CLIP | 99.0 | 37.4 | 63.5 | 99.9 | 82.1 | 70.3 | 99.6 | 54.0 | 89.5 | 87.8 | 52.7 | 38.5 | 72.9 |
| FatFormer | 99.0 | 91.4 | 46.0 | 100.0 | 75.8 | 72.6 | 99.7 | 82.0 | 99.9 | 99.9 | 83.2 | 50.7 | 83.4 |
| D3 | 98.6 | 51.0 | 100.0 | 99.8 | 99.9 | 85.3 | 97.8 | 90.2 | 90.2 | 99.7 | 92.1 | 99.8 | 92.0 |
| DIRE+AG | 100.0 | 86.2 | 93.8 | 76.5 | 99.8 | 90.2 | 89.9 | 96.0 | 98.7 | 71.2 | 92.5 | 90.9 | 90.5 |
| ConvNeXt | 100.0 | 99.0 | 100.0 | 99.0 | 100.0 | 99.0 | 100.0 | 74.0 | 100.0 | 89.4 | 89.8 | 100.0 | 95.9 |
| AG+ConvNeXt | 99.9 | 97.0 | 100.0 | 98.9 | 99.9 | 97.4 | 99.9 | 80.9 | 98.0 | 97.0 | 97.7 | 99.5 | 97.2 |
| ViT | 90.1 | 3.2 | 87.8 | 59.6 | 94.3 | 36.7 | 73.8 | 0.0 | 0.0 | 64.1 | 27.0 | 41.5 | 48.2 |
| AG+ViT | 85.1 | 24.6 | 82.4 | 44.5 | 88.2 | 65.1 | 79.4 | 39.0 | 86.4 | 49.9 | 57.2 | 41.8 | 62.0 |
| OG | 99.9 | 34.6 | 99.7 | 93.1 | 99.8 | 54.3 | 99.3 | 0.0 | 99.8 | 91.0 | 29.4 | 87.9 | 74.1 |
| AG (ours) | 100.0 | 99.6 | 99.9 | 99.9 | 99.9 | 99.9 | 99.9 | 98.0 | 100.0 | 99.9 | 99.9 | 99.4 | 99.7 |

Table 7: Cross-generator TPR@5%FPR (%) on DiffusionForensics.

| Method | ADM | BigGAN | Glide | Midjourney | SD1.4 | SD1.5 | VQDM | Wukong | Mean |
|---|---|---|---|---|---|---|---|---|---|
| ResNet50 | 47.9 | 47.2 | 47.1 | 48.1 | 46.5 | 46.6 | 51.2 | 41.1 | 47.0 |
| DIRE | 55.5 | 55.0 | 55.3 | 53.4 | 54.6 | 54.8 | 54.1 | 53.6 | 54.5 |
| NPR | 50.0 | 50.0 | 50.0 | 50.0 | 50.0 | 50.0 | 50.0 | 50.0 | 50.0 |
| DE-FAKE | 63.6 | 72.7 | 72.7 | 52.6 | 51.2 | 51.2 | 61.2 | 50.2 | 59.4 |
| DIRE+AG | 63.8 | 46.0 | 59.1 | 59.5 | 61.0 | 60.7 | 60.6 | 59.2 | 58.7 |
| UniFD | 77.7 | 69.0 | 52.2 | 52.5 | 60.5 | 50.5 | 63.9 | 60.1 | 60.8 |
| C2P-CLIP | 68.8 | 77.4 | 63.9 | 55.9 | 56.8 | 54.1 | 53.3 | 53.1 | 60.4 |
| FatFormer | 50.4 | 49.7 | 51.1 | 50.3 | 50.0 | 49.3 | 49.7 | 50.7 | 50.1 |
| D3 | 79.0 | 62.0 | 71.2 | 46.2 | 60.8 | 60.5 | 59.4 | 58.3 | 62.2 |
| ConvNeXt | 70.2 | 56.3 | 68.2 | 57.5 | 57.9 | 57.7 | 54.7 | 55.1 | 59.7 |
| AG+ConvNeXt | 75.3 | 55.5 | 74.9 | 58.4 | 51.1 | 51.0 | 66.3 | 51.3 | 60.5 |
| ViT | 51.8 | 60.9 | 57.6 | 54.0 | 50.2 | 50.8 | 59.0 | 48.3 | 54.1 |
| AG+ViT | 51.8 | 65.2 | 71.6 | 56.7 | 49.5 | 50.2 | 63.0 | 46.7 | 56.8 |
| OG | 64.3 | 74.4 | 87.3 | 54.8 | 52.1 | 51.8 | 63.4 | 49.8 | 62.2 |
| AG (Ours) | 65.8 | 60.3 | 73.3 | 64.3 | 60.5 | 60.6 | 59.8 | 54.9 | 62.4 |

Table 8: GenImage cross-dataset results (ACC, %) for all generators.

| Method | ADM | BigGAN | Glide | Midjourney | SD1.4 | SD1.5 | VQDM | Wukong | Mean |
|---|---|---|---|---|---|---|---|---|---|
| ResNet50 | 44.1 | 40.7 | 41.1 | 45.0 | 40.8 | 40.6 | 51.2 | 41.7 | 43.1 |
| DIRE | 87.6 | 72.2 | 78.4 | 70.6 | 75.8 | 76.0 | 68.9 | 70.1 | 75.0 |
| NPR | 36.3 | 37.7 | 37.2 | 46.8 | 58.9 | 59.1 | 45.9 | 55.1 | 47.1 |
| DE-FAKE | 71.8 | 82.3 | 82.7 | 57.6 | 53.9 | 53.5 | 69.0 | 52.2 | 65.4 |
| DIRE+AG | 74.6 | 45.8 | 74.2 | 71.4 | 68.8 | 68.6 | 66.3 | 68.1 | 67.2 |
| UniFD | 85.2 | 79.2 | 55.7 | 48.7 | 66.9 | 56.4 | 72.7 | 67.7 | 66.5 |
| C2P-CLIP | 86.9 | 87.4 | 72.6 | 49.7 | 66.1 | 64.9 | 61.8 | 59.3 | 68.6 |
| FatFormer | 50.2 | 50.0 | 50.5 | 50.2 | 50.1 | 49.6 | 49.6 | 50.6 | 50.1 |
| D3 | 87.7 | 76.6 | 80.0 | 48.5 | 66.2 | 65.7 | 76.3 | 63.4 | 70.6 |
| ConvNeXt | 83.6 | 73.6 | 82.5 | 72.8 | 69.9 | 77.7 | 74.3 | 61.3 | 74.5 |
| AG+ConvNeXt | 98.2 | 73.6 | 89.0 | 78.5 | 63.0 | 63.2 | 85.8 | 62.3 | 76.7 |
| ViT | 57.4 | 73.1 | 67.4 | 59.6 | 52.7 | 53.8 | 67.7 | 47.1 | 59.8 |
| AG+ViT | 53.4 | 78.5 | 79.7 | 59.4 | 49.6 | 50.5 | 67.0 | 45.3 | 60.4 |
| OG | 77.7 | 87.9 | 94.9 | 60.9 | 57.4 | 57.4 | 76.1 | 51.3 | 70.4 |
| AG (Ours) | 84.4 | 77.8 | 87.3 | 78.6 | 74.2 | 74.2 | 76.0 | 66.2 | 77.3 |

Table 9: GenImage cross-dataset results (AP, %) for all generators.

| Method | ADM | BigGAN | Glide | Midjourney | SD1.4 | SD1.5 | VQDM | Wukong | Mean |
|---|---|---|---|---|---|---|---|---|---|
| ResNet50 | 1.6 | 0.4 | 0.5 | 1.6 | 0.0 | 0.1 | 6.5 | 0.1 | 1.3 |
| DIRE | 48.3 | 22.1 | 25.2 | 12.0 | 21.6 | 23.5 | 15.7 | 12.8 | 22.7 |
| NPR | 0.9 | 0.0 | 0.4 | 2.9 | 11.6 | 11.5 | 5.0 | 11.3 | 5.5 |
| DE-FAKE | 27.5 | 44.0 | 42.2 | 8.3 | 6.2 | 6.3 | 23.4 | 5.5 | 20.4 |
| DIRE+AG | 0.0 | 0.0 | 0.0 | 0.0 | 0.0 | 0.0 | 0.0 | 0.0 | 0.0 |
| UniFD | 44.2 | 76.6 | 4.2 | 9.5 | 16.8 | 15.1 | 35.9 | 29.4 | 29.0 |
| C2P-CLIP | 47.2 | 65.5 | 55.6 | 1.7 | 19.1 | 17.0 | 31.8 | 10.2 | 31.0 |
| FatFormer | 1.8 | 1.8 | 1.8 | 1.7 | 1.3 | 1.5 | 1.7 | 1.4 | 1.6 |
| D3 | 52.1 | 51.8 | 47.3 | 10.4 | 18.6 | 17.0 | 21.2 | 15.8 | 29.3 |
| ConvNeXt | 40.7 | 24.2 | 43.0 | 22.3 | 19.6 | 18.2 | 15.7 | 10.9 | 24.3 |
| AG+ConvNeXt | 70.7 | 26.9 | 64.7 | 18.2 | 13.9 | 14.9 | 30.0 | 13.2 | 31.6 |
| ViT | 5.7 | 25.9 | 29.4 | 10.1 | 4.9 | 5.7 | 17.8 | 2.6 | 12.8 |
| AG+ViT | 13.8 | 28.1 | 32.9 | 9.5 | 3.4 | 4.1 | 15.1 | 2.4 | 13.7 |
| OG | 30.4 | 47.3 | 76.3 | 14.2 | 8.6 | 8.4 | 25.1 | 4.9 | 26.9 |
| AG (Ours) | 42.7 | 31.9 | 55.4 | 36.6 | 28.6 | 29.1 | 28.5 | 16.9 | 33.7 |

Table 10: GenImage cross-dataset results (TPR@5%FPR, %) for all generators.

