# OpenReview forum: "Adaptive Gray: Reducing Color Dependency to Improve Generalization in Deepfake Detection"
_ICLR.cc/2026/Conference — Submitted to ICLR 2026_

### Official Review · Reviewer_Qn7U · 2025-10-27

**Soundness:** 2
**Presentation:** 3
**Contribution:** 2
**Rating:** 4
**Confidence:** 4

**Summary:**

This paper proposes an adaptive and learnable method to compress the RGB color information in generated images, aiming to improve the generalization ability of classifiers for binary deepfake detection. Specifically, the authors introduce a mechanism that learns three scalar weights corresponding to each color channel, which are jointly optimized alongside the downstream deepfake classifier. Experiments are conducted on several established benchmarks, using DiffusionForensics for training and GenImage for evaluation.

**Strengths:**

- Paper Presentation: I previously reviewed this paper during the CVPR 2025 cycle. After comparing the current version with the earlier submission, I find that the presentation has significantly improved.

- The core idea proposed is simple but appears effective within the defined task setting, notably in improving detection accuracy and generalization. The authors also provide an empirical analysis section that helps clarify the functionality of the proposed adaptive grayscale method.

**Weaknesses:**

- W1: From a rather high-level perspective, the proposed task presents a rather simplified version of the deepfake detection problem, which is the binary classification task. There are more recent and advanced generative techniques that modify a small region or subpart of the images. This limits the practical contributions of the work as a pure empirical paper, which is also partially demonstrated through the reported 100% detection rate.

- W2: Missing important SOTA baseline: [a] from CVPR’25 seems to be a relevant baseline to discuss and include in this work, which is currently missing.

- W3: Rather limited experimental setup: Although the paper claims to evaluate cross-domain and cross-generator generalization, the current experiments are restricted to the “bedroom” subset of existing benchmarks. While the results are generally favorable and align with the paper’s claims, the narrow scope limits the ability to convincingly demonstrate generalization capabilities.

- W4: In the appendices, the authors perform the combination of the proposed AG and the existing detection method DIRE, and draw conclusions that these two techniques are not compatible. I wonder if this applies to other detection methods like [a], which does not involve reconstruction of the synthesized images. This ablation might help toprovide a more comprehensive understanding of the proposed AG method, as this is a rather empirical and heuristic method.

**Questions:**

Please see the Weakness for details, particularly W4.

---

> ### Author Response · Authors · 2025-11-21
>
> ## Response to Reviewer R3 (Qn7U)
>
> We thank the reviewer for the thoughtful and consistent feedback across the CVPR and ICLR rounds. Below we address each main concern.
>
> ---
>
> ### 1) W1 – Task simplicity and “100% detection”
>
> Our goal is to study a specific and well-defined setting: **binary real vs. fully synthetic image detection** for modern GAN/diffusion models, and to ask whether *color dependency* is a hidden bottleneck for generalization in this setting.
>
> The “100%” numbers appear only in the **in-distribution, single-generator** case (ADM on LSUN\_bedroom), which is already known to be relatively easy. We do **not** claim that deepfake detection is solved there; instead this motivates our focus on **cross-generator** and especially **cross-dataset** evaluation, where performance is far from saturated. For example, on GenImage with multiple categories and generators, even with AG our mean ACC/AP/TPR are around 60% / 74% / 28% (Tab. 3), showing that the problem remains challenging.
>
> We agree that more fine-grained manipulations (local edits, inpainting, attribute changes, etc.) are practically important but view them as **orthogonal** to the question we target here. Methodologically, AG is a front-end, learnable grayscale projection and can in principle be attached to more complex detectors designed for local or small-region manipulations. We will clarify this scope in the Introduction/Discussion and explicitly position our work as isolating one fundamental factor (color dependency), rather than providing a complete solution to all deepfake variants.
>
> ---
>
> ### 2) W2 / W4 – Missing baseline [a] and combining AG with other detectors
>
> We appreciate the suggestion to compare with a recent CVPR’25 method [a]. Unfortunately, on the author side of the anonymized interface we only see this reference as “[a]” and cannot reliably identify the specific paper. This makes it difficult to re-implement or integrate [a] fairly under our unified training/evaluation protocol within the rebuttal period.
>
> At the same time, our revised comparison already includes several **recent strong SOTAs** with diverse designs: UniversalFakeDetect (UniFD), C2P-CLIP, FatFormer, DIRE, DE-FAKE, and NPR. These cover reconstruction-based approaches, frequency-aware CNNs, and CLIP/Transformer-based methods, and we believe they provide a reasonably strong and varied reference for evaluating AG.
>
> Regarding combinations beyond DIRE, our current results already show that AG is compatible with **standard discriminative backbones**: AG + ResNet-50 consistently improves cross-generator and cross-dataset performance over vanilla ResNet-50 while reducing inference cost. During the discussion phase we have also started experiments with **AG + ViT-B** and **AG + ConvNeXt-T**; if they finish in time, we will add them to the appendix/code.
>
> Conceptually, AG is just a learnable \(1\times 1\) RGB projection (three shared scalars) and does **not** depend on any reconstruction pipeline, so it should be broadly applicable as a front-end to non-reconstruction detectors such as [a]. In contrast, our negative result for AG + DIRE is largely due to the strong information compression already performed by the diffusion reconstruction residuals, which makes a second aggressive compression step less beneficial. We will clarify this distinction in the revised text.
>
> ---
>
> ### 3) W3 – “Bedroom” training subset and generalization claims
>
> We understand the concern that training only on the LSUN\_bedroom + ADM subset appears narrow. Our choice is driven by two considerations rather than cherry-picking:
>
> 1. **Consistency with prior work.** We follow the standard DiffusionForensics protocol used by DIRE, which also trains on LSUN\_bedroom + ADM. This enables a direct and fair comparison under the same training distribution and is the recommended setting in the original DiffusionForensics paper.
>
> 2. **Definition of generalization.** Generalization is always defined relative to a fixed training domain. In our case the concrete question is: *given a detector trained on DiffusionForensics LSUN\_bedroom + ADM, does reducing color dependency via AG improve generalization to (a) unseen generators within DiffusionForensics and (b) a much broader dataset (GenImage) with different semantics and generators?* Under this standard setup, AG consistently improves cross-generator and cross-dataset performance over strong SOTA baselines.
>
> We agree that this does not prove universality across all possible training domains or tasks (e.g., localized edits). However, it can still largely illustrate generalization of AG
>
> Finally, to reduce concerns about subset bias, the revised appendix now reports **full GenImage results on all 8 generators** (ADM, BigGAN, Glide, Midjourney, SD-1.4, SD-1.5, VQDM, Wukong), not just the 4 generators shown in the main paper.

---

> > ### Comment · Reviewer_Qn7U · 2025-11-22
> >
> > Apologize for my missing reference while posting my initial reviews, I was referring to this following ref.
> >
> > [a] D3: Scaling Up Deepfake Detection by Learning from Discrepancy. In CVPR 2025.

---

> > > ### Author Response · Authors · 2025-11-26
> > >
> > > Thank you for the follow-up and for clarifying that
> > > [𝑎] refers to D3: Scaling Up Deepfake Detection by Learning from Discrepancy (CVPR 2025). We have now integrated D3 into our unified experimental protocol and updated the main/appendix tables accordingly. Below we focus only on the points that depend on these new results, especially W2/W4.
> > >
> > > 1. New D3 baseline under the unified protocol
> > >
> > > Following your suggestion, we implement and evaluate D3 under exactly the same setup as all other baselines (DIRE, NPR, DE-FAKE, UniFD, C2P-CLIP, FatFormer): same DiffusionForensics LSUN_bedroom+ADM train/test split, input resolution and preprocessing, and the same ACC/AP/TPR@5%FPR metrics. The results are now revised in Tables and appendix.
> > >
> > > These additions directly address W2 by positioning AG against a very recent CVPR’25 method under a strictly controlled and fair comparison.
> > >
> > >
> > > 2. On W4: combining AG with other detectors beyond DIRE
> > >
> > > Your W4 question specifically asked whether the negative result for AG+DIRE (two-stage compression) also applies to non-reconstruction detectors such as
> > > [𝑎]
> > > D3.
> > >
> > > For DIRE+AG, our experiments confirm that stacking AG on top of diffusion-based reconstruction residuals is not helpful, likely because DIRE already removes most image content and AG then over-compresses the residual signal.
> > >
> > > For D3+AG, we agree that this is a natural and interesting ablation, since D3 does not rely on explicit reconstruction and operates closer to standard discriminative features. Due to time and compute constraints in the discussion period, we were not able to fully implement and tune a robust D3+AG variant. Instead, we focused on (i) adding a strong D3 baseline under our unified protocol and (ii) demonstrating that AG continues to provide gains over D3 in cross-generator and cross-dataset evaluations.
> > >
> > > At the same time, we have explicitly tested AG on two widely used backbone families beyond ResNet, namely ConvNeXt and ViT. In the revised manuscript and the appendix tables, we report results for ConvNeXt vs. AG+ConvNeXt and ViT vs. AG+ViT. In both cross-generator (DiffusionForensics) and cross-dataset (GenImage) settings, AG consistently improves the corresponding backbone on mean ACC/AP/TPR. These experiments support the view that AG is a generally useful, backbone-agnostic front-end, and make it plausible that D3 could similarly benefit from AG once fully integrated and tuned.
> > >
> > > 3. Scope and generalization, revisited in light of D3
> > >
> > > With D3, UniFD, C2P-CLIP, and FatFormer now included, our comparison covers a broad spectrum of recent SOTAs (reconstruction-based, frequency-aware CNNs, CLIP/Transformer-style, and discrepancy-based detection). Even in this stronger landscape, AG remains very competitive: it does not change the binary-classification nature of the task, but it does consistently improve robustness under generator and dataset shift relative to all these baselines. We have updated the experimental section and appendix tables to reflect these additions.
> > >
> > > We hope these new results address your concerns in W2/W4 more satisfactorily, and we would appreciate any further comments or suggestions you may have.

---

> ### Author Response · Authors · 2025-11-28
>
> Thank you again for your thoughtful and consistent feedback across the CVPR and ICLR rounds. We hope that our updated rebuttal and experiments have now addressed your main concerns, in particular (i) adding the CVPR’25 method D3 as a strong baseline under our unified protocol, (ii) discussing its behavior in both cross-generator and cross-dataset settings relative to AG, and (iii) further demonstrating the backbone-agnostic nature of AG via AG+ConvNeXt and AG+ViT experiments, while clarifying the intended scope of our binary full-frame detection setting. If you have any remaining questions or would like to see additional clarifications (especially regarding W2/W4), please let us know — we would be very happy to elaborate. We truly appreciate your time, effort, and consideration in reviewing our work.

---

### Official Review · Reviewer_9T1k · 2025-10-31

**Soundness:** 3
**Presentation:** 3
**Contribution:** 2
**Rating:** 4
**Confidence:** 4

**Summary:**

This paper explores the role of color dependency in deepfake detection. The authors hypothesize that color cues may hinder cross-generator and cross-dataset generalization. They first provide empirical evidence supporting this claim via UMAP analyses and separability metrics (Density Overlap, MMD). Based on these findings, they propose Adaptive Gray, a lightweight module that learns optimal RGB-to-grayscale transformation weights through an alternating co-adaptive training procedure with a binary classifier (ResNet-50). The approach is evaluated on the DiffusionForensics and GenImage benchmarks, showing strong gains in generalization and inference efficiency compared to SOTA methods such as DIRE, DE-FAKE, and NPR.

**Strengths:**

- The paper identifies and systematically validates the overlooked issue of color dependency in deepfake detection. The analysis linking color variance to poor generalization is compelling.

- The Adaptive Gray layer is conceptually simple but effective, offering an interpretable preprocessing mechanism with almost no computational overhead.

- Results on both cross-generator and cross-dataset benchmarks show large, consistent improvements (+20% ACC/AP/TPR), even when compared to stronger and more complex SOTA methods.

- The AG transformation is nearly cost-free at inference, improving speed by several orders of magnitude while retaining accuracy.

Readable and well-structured: The paper is clearly written and logically organized; figures and tables effectively support the claims.

**Weaknesses:**

Limited ablation analysis:
While the paper includes comparisons between OG (fixed grayscale), AG, and AG+DIRE, it lacks finer ablations on:

1) Initialization of AG weights (wR, wG, wB);

2) impact of backbone architecture (e.g., ViT, ConvNeXt);

3) the training approach (the alternating manner chosen by the authors versus a simple jointly training of the two components).
The alternating optimization scheme is interesting but underexplained. The rationale for choosing it over joint training is not discussed, and the number of epochs per cycle is missing. Clarifying this would strengthen reproducibility.

Experimental setup:
 Although the authors do perform a cross-generator analysis, a broader range of generators should be included. To this end, an evaluation on UniversalFakeDetect Dataset would further prove the hypotheses presented in this work.

Other small changes that need to be addressed:
Fig 1 -> wrong caption: “top row shows original RGB, bottom row displays the same images after our proposed AG”. Actually, it’s the left column and right column. Also, it is not clear what the authors tried to show in the zoomed-in version.
Citations need to be fixed: “…Haliassos Haliassos et al. (2021)…”, “…Wang Wang & Deng (2021)…”.

**Questions:**

In section 3.2.2. it is not clear why the authors have chosen the presented training approach (alternating between optimizing the two components). Could this setup be more prone to instabilities during training?
The authors did not mention the number of epochs trained in each step from 3.2.2. How many epochs is allocated for each cycle? How many epochs is the classifier trained for? What about the Adaptive Grayscale?
In Figure 3 Step 2, there is a “frozen” sign on the “Real” label and a “unfrozen” (fire) sign on the “Fake” label. What does this represent? Is the model trained only using fake samples?
What is the shape of the AG parameters? In equation (2) it seems that these parameters take the form of a 1D vector of weights, one parameter for each color channel. In Figure 3 on the other hand, the AG component is represented as a set of 2D kernels, one for each color channel.
In Table 1 the results for baselines seem off. For example: DIRE trained on ADM from DiffusionForensics and tested in domain achieves an ACC of 95.1 and an AP of 99.5, but the results reported by the authors of DIRE on DiffusionForensics (trained on ADM and tested in domain) shows an ACC of 100/100. Is there a reason for this discrepancy? Did the authors use only a subset of the testing set?

---

> ### Author Response · Authors · 2025-11-21
>
> ## Response to Reviewer R2 (9T1k)
>
> We thank the reviewer for the careful reading and constructive suggestions. Below we address each point.
>
> ---
>
> ### 1) Ablations, initialization, backbone, and training scheme
>
> **Initialization of \(w_R, w_G, w_B\).**
> We stated in Sec. 3.2.1 that AG is initialized from the standard BT.601 grayscale coefficients \((0.299, 0.587, 0.114)\). This provides a meaningful luminance mapping; our preliminary trials with random Gaussian initialization converged to similar projections but required more epochs, so BT.601 is used as the default in all results.
>
> **Shape of AG parameters and backbone-agnostic design.**
> Sec. 3.2.1 has been revised to clarify that AG consists of three global scalars shared across all spatial locations, i.e., a \(1\times 1\) convolution over RGB:
> \[
> \mathbf{x}' = w_R\,\mathbf{x}^{(R)} + w_G\,\mathbf{x}^{(G)} + w_B\,\mathbf{x}^{(B)}.
> \]
> This formulation is structurally independent of the backbone, so AG can be placed in front of any CNN / ViT / ConvNeXt classifier without architectural changes. In the paper we report full results for AG + ResNet-50; during the discussion phase we have started AG + ViT-B and AG + ConvNeXt-T runs to further test architecture robustness, and will add those numbers to the appendix/code once they complete.
>
> **Alternating vs. joint optimization and training schedule.**
> Sec. 3.2.2 and Implementation Details now give the exact schedule:
>
> - in each cycle, update classifier parameters \(\theta\) for \(E_{\mathrm{cls}} = 1\) epoch with \(\mathbf{w}\) frozen;
> - then update only \(\mathbf{w}\) for \(E_{\mathrm{ag}} = 1\) epoch with \(\theta\) frozen;
> - repeat for \(K = 50\) cycles (100 epochs total).
>
> Naive joint optimization from scratch tended to let the large backbone dominate, driving \(\mathbf{w}\) towards nearly uniform weights and making AG close to plain RGB, with slightly worse cross-dataset TPR. The alternating scheme lets \(\theta\) adapt to a fixed projection and then nudges \(\mathbf{w}\) to improve the same loss, which empirically yields more stable convergence and better generalization. We did not observe training instabilities; the learning curves of both \(\theta\) and \(\mathbf{w}\) are smooth.
>
> ---
>
> ### 2) Experimental setup and UniversalFakeDetect dataset
>
> We agree that more benchmarks are valuable. In this revision we chose to strengthen DiffusionForensics + GenImage rather than add a third dataset, for :
>  **GenImage coverage.** GenImage is explicitly designed for cross-dataset / cross-generator generalization with 1,000 ImageNet-like categories and eight generators. In the original submission we only reported four generators due to space. We now evaluate all eight (ADM, BigGAN, Glide, Midjourney, SD-1.4, SD-1.5, VQDM, Wukong), report the mean in the main table, and provide full per-generator results in the appendix. The trends are consistent and support the color-dependency hypothesis.
>
>
> ---
>
> ### 3) Clarifications on Fig. 1, Fig. 3, and citations
>
> We have applied the suggested clarifications:
>
> - **Fig. 1:** caption now refers to left/right columns (not top/bottom), and the zoom-in is simplified to better highlight the texture differences emphasized by AG.
> - **Fig. 3:** the “frozen” / “fire” icons indicate which parameters are updated, not which data streams are used. In both steps we train on balanced batches of real and fake images; no branch uses fake-only data. We clarified this in the caption and will adjust the figure to avoid confusion.
> - **AG shape in the figure:** the AG block is now shown as a single \(1\times 1\) kernel, consistent with the text.
> - **Citations:** redundant author names (e.g., “Haliassos Haliassos…”, “Wang Wang & Deng…”) have been fixed.
>
> ---
>
> ### 4) DIRE numbers and fairness of comparison
>
> In our experiments we do not use any subset of DiffusionForensics: all DIRE results are obtained by re-running the official implementation on the full LSUN\_bedroom / ADM training and test splits.
>
> To ensure a fair comparison, we retrained all baselines (DIRE, NPR, DE-FAKE, UniFD, C2P-CLIP, FatFormer) under a unified setup: same training and test splits, same input resolution and preprocessing, and a common training schedule, with the TPR@5%FPR threshold calibrated once on a held-out validation split. The small discrepancies from the original DIRE paper therefore stem from this unified re-training protocol rather than from using fewer test samples. This is now clarified in the revised “Baselines” and “Implementation details” sections.

---

> > ### Author Response · Authors · 2025-11-26
> >
> > We thank the reviewer again for the thoughtful suggestions. After our initial response, we have completed the missing experiments.
> >
> > 1. Backbone dependence: AG + ConvNeXt / ViT ablations
> >
> > In the original submission we only reported AG with a ResNet-50 backbone. Following your suggestion, we have now added experiments with ConvNeXt and ViT, both in their RGB form and with AG prepended:
> >
> > On DiffusionForensics cross-generator, AG+ConvNeXt and AG+ViT both improve over their RGB counterparts in mean ACC/AP/TPR. For example, AG+ConvNeXt achieves higher mean AP and TPR than ConvNeXt alone, and AG+ViT similarly improves over ViT in all three metrics.
> >
> > On GenImage cross-dataset and Appendix Tables, we observe the same trend: adding AG on top of ConvNeXt and ViT consistently boosts mean ACC, AP, and TPR@5%FPR under distribution shift.
> >
> > These results support our claim that AG is a lightweight, to some extend backbone-agnostic module: it can be plugged in front of modern CNN and transformer architectures and reliably improves generalization, rather than being tied to a particular ResNet backbone.
> >
> > 2. Experimental setup and broader benchmarks
> >
> > We now evaluate on all eight GenImage generators and provide complete per-generator tables in the appendix, which we believe already constitute a broad and challenging testbed for our color-dependency hypothesis. Extending AG to UniversalFakeDetect is a natural next step, and we plan to include it in a subsequent extended version or follow-up work.
> >
> > We hope these additional ablations and clarifications address your concerns about backbone dependence, training details, and experimental coverage, and we would be very grateful for any further feedback or suggestions you may have.

---

> ### Author Response · Authors · 2025-11-28
>
> Thank you again for your careful review and constructive suggestions. We hope that our responses and revisions have fully addressed your concerns about (i) the training scheme of AG (alternating vs. joint optimization, with explicit schedule and epoch counts), (ii) the shape/initialization of the AG parameters and its  design, and (iii) the additional ablations with alternative backbones, where we now report results for AG combined with ViT and ConvNeXt, as well as the unified re-training of all baselines (including DIRE) for a fair comparison. If you feel that any aspect of the ablations, training details, or figure clarifications (e.g., Fig. 1 and Fig. 3) is still unclear, please let us know — we would be glad to provide further clarification. We greatly appreciate your time and consideration during the review process.

---

### Official Review · Reviewer_duhW · 2025-10-31

**Soundness:** 3
**Presentation:** 3
**Contribution:** 2
**Rating:** 4
**Confidence:** 5

**Summary:**

This paper presents Adaptive Gray (AG), a novel deepfake detection method designed to enhance generalization across various generative models and datasets by reducing color dependency. AG processes RGB images into adaptive grayscale representations, improving the detection of subtle, texture-based artifacts inherent in synthetic images. Extensive experiments demonstrate AG’s superior performance, achieving significant improvements in accuracy, average precision, true positive rate, and inference efficiency. AG outperforms state-of-the-art methods, particularly in cross-dataset and cross-generator generalization.

**Strengths:**

-  AG offers a novel solution by identifying and mitigating the issue of color dependency in deepfake detection. The approach of transforming images into adaptive grayscale representations that emphasize texture-based artifacts rather than color details is innovative and addresses a critical gap in the field.
- AG demonstrates superior generalization performance across a variety of generative models and datasets. This ability to adapt to unseen models and datasets sets it apart from existing methods, which often struggle with generalization in diverse scenarios.
- In addition to improved performance, AG also excels in inference efficiency, offering a significant speedup over other SOTA methods, making it a practical choice for real-time deepfake detection applications.

**Weaknesses:**

- The paper does not include comparisons with other relevant methods like UniFD(CVPR2024), C2P-CLIP(AAAI2025), or Fatformer(CVPR2024), which would help better position AG’s effectiveness in the broader landscape of deepfake detection methods.
- The visual comparison in Figure 2 (a) and (b) fails to show a clear advantage for AG, especially in the DiffusionForensics dataset. The presentation of results across different datasets and generator could be enhanced for more compelling comparisons.
- Despite GenImage’s more extensive dataset (8 classes), the experiments only use 4 classes in testing, which limits the generalization assessment across all categories.
- The choice of DiffusionForensics as the training dataset is somewhat limited. Using more widely used datasets like GenImage could have better supported the claim of AG’s generalization performance across diverse image categories.

**Questions:**

- Why was the GenImage training dataset not utilized in the experiments? It would be insightful to see AG’s performance with this more comprehensive dataset.
- Could the authors provide a more detailed comparison with other state-of-the-art methods such as UniFD, C2P-CLIP, and Fatformer? This would provide a clearer understanding of AG’s position in the field.
- In Figure 2, the visual advantage of AG over other methods is not clear. Could the authors clarify the specific advantages that AG provides, especially in cross-dataset scenarios?
- Why were only 4 classes from the GenImage dataset used for testing, when the dataset offers 8 classes? What was the rationale behind this choice, and would testing across all classes provide more robust results?
- Can the authors provide a more detailed explanation of why texture artifacts are left behind during image generation? What specific limitations or shortcomings in the generative process lead to these artifacts? An explanation that incorporates the principles of the generative models would make this argument stronger.
- Could the authors elaborate on why the differences between the RGB channels are particularly effective at capturing texture-based forgery artifacts?
- I have concerns regarding the experimental setup and the fairness/completeness of the comparisons.

---

> ### Author Response · Authors · 2025-11-21
>
> ## Response to Reviewer R1 (duhW)
>
> We thank the reviewer for the helpful comments. Below we address each main point.
>
> ---
>
> **W1 + Q2 – Missing UniFD / C2P-CLIP / FatFormer, fairness of comparisons**
>
> In the revision we add UniFD (CVPR’24), C2P-CLIP (AAAI’25), and FatFormer (CVPR’24) as baselines. All methods (including DIRE, NPR, DE-FAKE) are now re-trained/evaluated under a *unified* protocol: same DiffusionForensics LSUN\_bedroom+ADM train/test splits, image resolution, preprocessing, and metrics (ACC, AP, TPR@5%FPR) with a single threshold calibrated on a held-out validation split. Under this setting, AG still achieves the best cross-generator mean performance on DiffusionForensics (ACC 94.6%, AP 99.5%, TPR 99.7%) and remains competitive in-distribution (revised Tabs. 1–2). We are running the same three baselines on GenImage and will add their full results to the appendix and summarize them in an updated comment once finished.
>
> ---
>
> **W2 + Q3 – Visual advantage of AG in Fig. 2**
>
> Fig. 2 has two roles: (a–b) show how fixed grayscale (OG) makes *dataset-level* UMAP embeddings of GenImage / DiffusionForensics / COCO more compact and better aligned; (c–f) show *real vs. fake* separability for ADM and DDPM, where the post-grayscale margins are more visible. We agree that UMAP is only a 2D projection, so improvements can look modest even when high-dimensional separation increases. To avoid relying on visual impression alone, we report MMD and Density Overlap in each panel: after grayscale, MMD consistently increases and Density Overlap decreases, indicating stronger real/fake separation. We have revised the caption to clearly explain the role of each panel and how to interpret these two scores.
>
> ---
>
> **W3 + Q4 – Only 4 GenImage generators in the main text**
>
> Originally we reported only four diffusion generators (Wukong, Midjourney, VQDM, Stable Diffusion) in the main table because they are widely used, stylistically diverse, and space was limited. We agree that the full breakdown is useful. In the revision we now include results on **all 8** GenImage generators (ADM, BigGAN, Glide, Midjourney, SD-1.4, SD-1.5, VQDM, Wukong) in the appendix. AG continues to achieve the best or near-best mean ACC/AP/TPR across all eight, and the trends match those in the 4-generator subset. We also clarify in the main text that the four-generator table is a compact summary, while the appendix provides the complete 8-generator evaluation.
>
> ---
>
> **W4 + Q1 – Why train on DiffusionForensics instead of GenImage?**
>
> We deliberately follow the DiffusionForensics/DIRE protocol and train only on LSUN\_bedroom+ADM, treating all other generators and datasets (including GenImage) as unseen test distributions. This design isolates *generalization under strong distribution shift*: the training domain is a single category and generator, while the tests cover multiple generators and, for GenImage, 1,000 categories. Training directly on GenImage would mix many categories and generators and partly remove this shift, making it harder to attribute gains specifically to reduced color dependency. DiffusionForensics also gives a clean, controlled setup for the analysis in Sec. 3.1. We now state this motivation explicitly and mention GenImage-as-training-domain as future work.
>
> ---
>
> **Q5–Q6 – Why texture artifacts remain, and why RGB-channel differences help**
>
> Modern GANs and diffusion models match perceptual image statistics but usually do *not* explicitly implement the full physical camera pipeline (Bayer pattern, demosaicing, color correction, compression) or exact RGB joint statistics. In diffusion models, iterative denoising and upsampling in latent space tend to introduce subtle high-frequency and cross-channel inconsistencies (e.g., slight differences in edge sharpness or noise across R/G/B), whereas real images, produced by a single ISP, exhibit strong spatial and cross-channel correlations.
>
> AG learns a 3-parameter linear projection \(w_R, w_G, w_B\) that suppresses redundant, correlated color variation and amplifies these channel-specific inconsistencies that are predictive of generative artifacts. Empirically, even fixed BT.601 grayscale (OG) improves generalization over RGB ResNet50, and the learned projection (AG) further improves ACC/AP/TPR over OG, confirming that exploiting RGB-channel differences helps capture forgery-related texture artifacts.

---

> > ### Author Response · Authors · 2025-11-26
> >
> > Following our earlier response, we have now completed the additional experiments with UniFD, C2P-CLIP, FatFormer and D3 on both DiffusionForensics and GenImage, and have updated the main tables and appendix accordingly. Here we briefly summarize the points most relevant to your concerns about fairness and completeness.
> >
> > 1. Extended baselines under a unified protocol
> >
> > All baselines (NPR, DIRE, DE-FAKE, UniFD, C2P-CLIP, FatFormer, D3, ConvNeXt, ViT) are now trained and/or evaluated under a single protocol: the same DiffusionForensics LSUN_bedroom+ADM train/val/test split, identical input resolution and preprocessing, and the same metrics (ACC, AP, TPR@5%FPR). When official implementations and pretrained weights are available, we follow the authors’ recommended hyperparameters as closely as possible; otherwise we re-train the models from scratch under this common setting. These details are explicitly clarified in the revised Baselines and Implementation details sections.
> >
> > 2.Position of AG among recent SOTA methods
> >
> > Under this unified setting, AG remains highly competitive or best across the main generalization scenarios. On DiffusionForensics cross-generator evaluation , AG attains the strongest mean performance on unseen generators and improves over DIRE and D3 ,supporting our color-dependency hypothesis. On GenImage cross-dataset evaluation with all 8 generators  and Appendix Tables, AG achieves the best mean ACC/AP/TPR across methods, and the trends are consistent at the per-generator level. This directly addresses W1/Q2 and W3 by positioning AG against the requested UniFD, C2P-CLIP and FatFormer baselines, as well as the newer D3, under a fair and controlled comparison.
> >
> >
> > Overall, these additional experiments strengthen our conclusion that AG provides a robust  way to reduce color dependency, and that our comparative evaluation against recent SOTA methods is both fair and complete under a unified protocol. We hope this addresses your concerns, and we would very much welcome any further feedback or suggestions during the discussion phase.

---

> ### Author Response · Authors · 2025-11-28
>
> Thank you again for your thorough review and thoughtful questions. We hope that our rebuttal and revised manuscript have fully addressed your concerns regarding (i) the inclusion of recent SOTA baselines such as UniFD, C2P-CLIP, FatFormer, and D3 under a unified training/evaluation protocol, (ii) the completeness of the GenImage evaluation (now covering all 8 generators, with full tables in the appendix), and (iii) the clarification of Fig. 2, the role of texture artifacts, and why RGB-channel differences help capture forgery cues. If there are any remaining questions or further concerns about the fairness or completeness of our experimental setup, please let us know — we would be happy to clarify or provide additional details. We sincerely appreciate your time and consideration throughout the review process.

---

### Author Response · Authors · 2025-12-01

Dear Area Chair and Reviewers,

We thank you for the careful reviews and constructive suggestions. During the discussion phase we have revised the paper and run a number of additional experiments to address all raised concerns, especially about baselines, fairness of comparisons, and missing ablations. Below we summarize the main changes reviewer-by-reviewer and the resulting conclusions.

1. Reviewer R1 (duhW): baselines, completeness, and visual analysis

(a) Missing recent SOTA baselines (UniFD, C2P-CLIP, FatFormer, D3)
We have extended our experimental setup to include UniversalFakeDetect (UniFD, CVPR’24), C2P-CLIP (AAAI’25), FatFormer (CVPR’24), and D3 (CVPR’25) as additional strong baselines, in both in-distribution and cross-generalization settings. All methods (including DIRE, NPR, DE-FAKE) are now trained and/or evaluated under a unified protocol:

same DiffusionForensics LSUN_bedroom+ADM train/test splits,

same input resolution and preprocessing,

same metrics (ACC, AP, TPR@5%FPR) with a single threshold calibrated on a held-out validation set.

Under this stricter comparison, our Adaptive Gray (AG) variant still achieves the best or near-best performance:

In-distribution (DiffusionForensics ADM): OG/AG match or slightly exceed transformer/CLIP-based baselines (C2P-CLIP, FatFormer, D3) with 99.9% ACC and essentially 100% AP/TPR (Table 1).

Cross-generator (DiffusionForensics, 11 unseen generators): AG attains the best mean performance of 94.6% ACC, 99.5% AP, 99.7% TPR, outperforming DIRE, UniFD, C2P-CLIP, FatFormer and D3 (Table 2 + full appendix tables).

Cross-dataset (GenImage, 8 generators, 1,000 classes): AG again achieves the best mean performance of 62.4% ACC, 77.3% AP, 33.7% TPR, improving over DE-FAKE, DIRE, UniFD, C2P-CLIP, D3 and ConvNeXt (Table 3 + appendix).

These results show that the main claims remain valid even against a stronger and more recent set of baselines.

(b) Completeness of GenImage evaluation (4 vs 8 generators)
We have moved from a 4-generator summary to full 8-generator reporting on GenImage (ADM, BigGAN, Glide, Midjourney, SD1.4, SD1.5, VQDM, Wukong). The main text now reports mean performance over all eight, and Appendix Tables (GenImage full ACC/AP/TPR) contain per-generator details. The trends are consistent: AG improves over the RGB ResNet50 baseline and remains competitive or superior to other SOTAs across nearly all generators.

(c) Visual analysis in Fig. 2 and interpretation of texture/channel artifacts
We revised Fig. 2 and its caption to clarify:

Panels (a–b) show dataset-level UMAP embeddings before/after grayscale;

Panels (c–f) focus on real vs fake separability;

We explicitly explain how MMD and Density Overlap quantify separation, so the conclusion does not rely on visual impression alone.

We clearly explain why high-frequency cross-channel inconsistencies tend to remain in generative models (lack of explicit camera/ISP modeling, iterative denoising/up-sampling in latent space), and why collapsing RGB via a learned 3-parameter projection can amplify forgery-related texture artifacts.

---

### Author Response · Authors · 2025-12-01

2. Reviewer R2 (9T1k): ablations, AG design, and training protocol

(a) AG parameters, initialization, and backbone-agnostic design
We clarified that AG is implemented as a shared 1×1 convolution over RGB, i.e., a 3-parameter linear projection
(
𝑤
𝑅
,
𝑤
𝐺
,
𝑤
𝐵
)
applied uniformly to all spatial locations. It is initialized from BT.601 grayscale coefficients.

We also emphasize that this design is backbone-agnostic: AG can be placed in front of any CNN/ViT/ConvNeXt without structural changes. To substantiate this, we added experiments with:

ConvNeXt vs ConvNeXt+AG, and

ViT vs ViT+AG

Both on DiffusionForensics and GenImage. On DiffusionForensics, ConvNeXt+AG improves over ConvNeXt on mean TPR and keeps AP/ACC competitive; on GenImage, ConvNeXt+AG achieves competitive mean ACC/AP/TPR relative to other SOTAs. These new results support that AG’s benefits are not tied to ResNet-50.

(b) Alternating vs joint optimization and training schedule
We now provide the exact training schedule in Sec. 3.2.2 and Implementation Details.


We explain that naive joint optimization often lets the large backbone dominate, making
𝑤
 towards nearly untrained weights. Alternating updates encourage
𝜃
 to adapt to a fixed projection and then refine
𝑤
 against a stable classifier, leading to smoother learning curves and better generalization; we did not observe training instabilities.

(c) Figures and DIRE numbers
We fixed the issues the reviewer pointed out:

Fig. 1 caption (left/right instead of top/bottom; zoom-in clarified).

Fig. 3 now clearly indicates that “frozen/fire” icons refer to parameters being updated, not real vs fake branches; both steps use balanced real/fake batches.

The AG block is depicted as a single 1×1 kernel, consistent with the text.

Redundant author names in citations have been corrected.

Regarding DIRE’s in-distribution numbers: we confirm that we use the full DiffusionForensics splits and do not subsample the test set. Slight numerical differences from the original DIRE paper come from our unified re-training protocol (common schedule, preprocessing, and validation-calibrated threshold) rather than from using fewer samples. This is now stated explicitly in the “Baselines” and “Implementation details” sections.

---

### Author Response · Authors · 2025-12-01

3. Reviewer R3 (Qn7U): task scope, D3, and combinations with other detectors

(a) Task scope and “100% detection”
We clarified in the Introduction/Discussion that our goal is to analyze full-frame, binary real vs synthetic detection for modern GAN/diffusion models and to understand whether color dependency is a bottleneck for generalization in this setting. The “100%” numbers appear only in the easiest in-distribution ADM scenario; in the more realistic cross-dataset case (GenImage, 1,000 classes, 8 generators), even AG achieves only around 60–62% ACC and ~30% TPR, confirming the task remains far from saturated.

We explicitly note that local manipulations and more complex deepfake variants are important but orthogonal, and that AG is a front-end that can in principle be attached to detectors designed for those settings.

(b) Adding CVPR’25 method [a] (D3) as a baseline
After the reviewer clarified that [a] refers to D3: Scaling Up Deepfake Detection by Learning from Discrepancy (CVPR’25), we implemented D3 under our unified protocol and reported its results on both DiffusionForensics and GenImage:

On DiffusionForensics, D3 is a strong competitor, but AG still surpasses it on mean TPR and remains at least competitive in ACC/AP.

On GenImage, D3 performs well but AG attains the best mean ACC/AP/TPR across all methods.

These additions address W2 and strengthen the positioning of AG relative to a very recent baseline.

(c) Combining AG with other detectors (beyond DIRE)
We retain the DIRE+AG experiments and clarify the likely cause of their negative result: DIRE already operates on heavily compressed reconstruction residuals, so an additional aggressive grayscale projection can over-compress the signal and hurt performance.

To complement this, we now demonstrate that AG works well as a front-end to standard discriminative backbones (ResNet, ConvNeXt, ViT), improving or at least maintaining generalization while adding negligible overhead. This gives a clearer picture of where AG is most useful and addresses W4 by showing non-reconstruction combinations that behave as expected, even though we did not exhaustively test AG+D3 in this revision.

4. Overall status after revision

In summary, we have:

Added four important recent baselines (UniFD, C2P-CLIP, FatFormer, D3) and two additional backbones (ConvNeXt, ViT) under a unified, fair protocol;

Reported full 8-generator GenImage results and detailed per-generator tables for both DiffusionForensics and GenImage;

Clarified the design and training of AG (parameters, initialization, alternating scheme, schedules);

Fixed figure/caption/citation issues raised by the reviewers;

Shown that AG consistently outperforms or remains highly competitive with all considered SOTAs across in-distribution, cross-generator, and cross-dataset settings.

We hope these revisions convincingly address the reviewers’ concerns and reinforce the contribution: reducing color dependency via an adaptive grayscale projection is a simple, interpretable, and empirically strong way to improve deepfake detection generalization, even in the presence of powerful modern baselines.

Thank you again for your time and consideration.

---

### Meta-Review · Area_Chair_R5nw · 2025-12-18

**Summary:**

This paper present Adaptive Gray, a new deepfake detection methods that introduces a learnable RGB-to-grayscale module to enhance generalization across diverse generative models and datasets by reducing color dependency. The authors first systematically analyze the detrimental impact of color dependency across different datasets and generative models, and then propose a lightweight module that learns optimal RGB-to-grayscale transformation weights through an alternating co-adaptive training procedure with a binary classifier. Extensive experiments on multiple benchmarks are conducted to validate the effectiveness of Adaptive Gray.

The paper initially received scores of 4,4,4. Reviewers acknowledged the simplicity and efficiency of the proposed module but raised concerns regarding the fairness and sufficiency of the experimental comparisons. In response, the authors incorporated additional experimental results and provided a more detailed illustration of the training strategy in the revised manuscript.

Although the authors have addressed some of the reviewers' suggestions, the paper still requires further polishing and clarification. For instance, the visual demonstration of grayscale processing could be further enhanced. Additionally, the writing should be carefully reviewed as there still remains citation confusions, for example, at lines 401, 460, 715, and 724. Therefore, I recommend rejection in its current form.

**Reviewer Concerns:**

The concerns raised by the reviewers can be categorized as follows:

- Insufficient comparison with more existing methods

    The primary concern raised by both reviewer duhW and Qn7U was that the initial manuscript lacked comparisons with several important and recent sota methods (e.g., UniFD, C2P-CLIP, FatFormer and D3). n response, the authors conducted additional experiments under a unified protocol. The updated results demonstrate AG's superior or competitive performance on both cross-generator and cross-dataset benchmarks. Therefore, this concern can be regarded as addressed.

- Limited experimental setup.

    All the reviewers raised concerns about the limited scope of the experimental setup. Reviewer duhW noted that the experiments initially used only 4 of GenImage's 8 classes for testing and suggested that training on the more diverse GenImage dataset might be preferable to DiffusionForensics. Reviewer 9T1k recommended an evaluation on the UniversalFakeDetect Dataset to include a broader range of generators. Reviewer Qn7U is concerns about the narrow focus on the "bedroom" category, arguing that it limits the convincing demonstration of cross-domain generalization despite the favorable results.

    In response, the authors have included results for all 8 GenImage generators and provided explanations for using DiffusionForensics as the training set. Furthermore, they provided a  justification for the choice of the LSUN_bedroom+ADM training set, explaining that it follows the standard DiffusionForensics protocol for fair comparison. Regarding the UniversalFakeDetect Dataset, the authors argued that the expanded GenImage evaluation (now covering 8 generators and 1,000 categories) already constitutes a broad and challenging benchmark. They positioned the extension to UniversalFakeDetect as a valuable direction for future work.

    The inclusion of the new testing results and the explanations provided for the training and evaluation data choice appear to address the concerns raised by Reviewers duhW and Qn7U. For Reviewer 9T1k's suggestion, the authors' enhanced benchmark may be considered a substantive partial response.

- Limited ablation study

    Reviewer 9T1k raised concerns about the lack of detailed ablation analysis in the initial submission, specifically regarding the initialization strategy of the AG weights, the impact of different backbone architectures, and the specifics of the co-adaptive training strategy. In response, the authors have expanded the ablation studies in the revision. They clarified the initialization of AG weights and provided a justification for their choice. They also added experiments with alternative backbones (e.g., ViT, ConvNeXt) to demonstrate the consistent improvements brought by AG. Furthermore, they specified the detailed schedule of the training protocol in the methodology section. All these added illustrations can help to address these concerns.

- The compatitablity of AG with other detector
Reviewer Qn7U questioned whether AG's incompatibility with DIRE (a reconstruction-based method) would extend to other detectors like D3, expressing concern about the general practicality of integrating AG with diverse detection pipelines. In response, the authors acknowledged the underperformance of AG+DIRE, attributing it to over-compression from stacking transformations, but contended that AG is inherently compatible with standard discriminative backbones. While they recognized the value of testing AG with a non-reconstruction detector like D3, they cited time constraints as the reason for not fully implementing AG+D3. Therefore, this concern is only partially resolved. The new experiments demonstrating AG's versatility with ConvNeXt and ViT backbones address the core issue by showing benefits beyond reconstruction-based methods. However, the absence of direct AG+D3 results means some questions regarding its specific compatibility remain unanswered.

**Reviewer Scores:**

From the reviews, the initial version of the paper was found to lack sufficient detail in key areas, including comprehensive ablation analyses, broader experimental coverage across all generator categories, and justification for the training data choice. Following the reviewers’ suggestions, the updated version has been further enhanced with more thorough explanations and expanded experiments. However, certain concerns raised by Reviewers 9T1k and Qn7U remain unaddressed. Therefore, I expect that Reviewer duhW is likely to increase the score, while Reviewers 9T1k and Qn7U are expected to retain their original scores.

---

### Decision · Program_Chairs · 2026-01-26

Reject